# HiViBiX: Hierarchical Visually-informed Mono-to-Binaural Music Generation using Ambisonics

## Abstract

Binaural audio, a specialised form of stereo sound, provides depth and spatial localisation for highly immersive listening experiences, making it fundamental in modern entertainment. Prior research has largely relied on visual cues to directly adapt mono signals into binaural or to estimate transfer functions that induce spatiality. In contrast, we introduce HiViBiX, a novel framework that redefines the music representation by predicting first-order Ambisonics channels, which explicitly control the spatial positioning of the audio components in the generated binaural signal. Unlike existing multimodal approaches that extract spatial cues exclusively from full-frame RGB images, HiViBiX incorporates a hierarchical visual encoder that jointly models local music sound sources and their spatial depth with global environmental context. This design enables richer multimodal grounding and more precise spatialization. Extensive experiments on three widely used musical benchmarks: FAIR-Play, Music-Stereo, and YT-Music demonstrate that HiViBiX establishes new state-of-the-art performance for mono-to-binaural generation. We also show that our method achieves good results in out-of-domain context, using a simple adapter. Samples are available in the following repository: https://hivibix.vercel.app.

## 1 Introduction

Immersive audio playback, where the spatial position of a sound source can be perceived solely through auditory cues, has become a cornerstone of modern media applications. In domains such as gaming, virtual and augmented reality, and cinematic production, spatial audio is not merely an aesthetic enhancement but a functional necessity. It enables dramaturgical control by directing user attention, articulating scale and distance, and conveying events that occur outside the immediate visual frame. Unlike conventional stereo, binaural audio recreates a perceptual sense of space that aligns more closely with natural human hearing, allowing two-channel playback systems to deliver an experience of depth, realism, and presence. As interactive and immersive technologies continue to grow in scale and impact, the demand for accurate and efficient methods of binaural audio generation is more pressing than ever.

Ambisonics (Zotter & Frank, 2019) represents a special class of format, extensively used across the audio and music industry by hardware manufacturers, broadcasting services, and streaming platforms. Beyond these technical domains, Ambisonics also play a central role in entertainment applications such as cinema, gaming, and virtual reality, where precise spatial rendering is essential for immersion and realism. Using this format, audio sources can be captured, stored and played in an arbitrary manner by relying on spherical harmonics encoding. We selected this format as it represents a stronger alternative to Head-Related Transfer Functions (HRTFs) or Impulse Responses (IRs) (Ratnarajah & Manocha, 2024) because it allows for energy-preserving rotations and more stable localisation in the high frequency domain. Another important advantage of the Ambisonics over classical HRTFs is that the latter are highly dependent on the ear anatomy of the listener, while Ambisonics are more generalizable, especially if higher-order are used.

Visual cues are also an indispensable tool in binaural generation, as they contain priors over both important aspects, such as source sounding object position or depth, but also over intrinsic scene-related features such as room reverberations or different occlusions. This phenomenon is closely

related to the multisensory integration mechanisms that govern over human hearing. As such, the visual stream is often used to create an implicit abstract visual-to-spatial mapping. Most previous methods have focused on obtaining this mapping solely from a single pre-trained model. Our approach focuses on extracting modality-specific priors, thereby enforcing more coherence between the available mono music audio signal and visual knowledge to help in rendering more realistic binaurals.

In this work, we present **HiViBiX**, a novel approach to image-conditioned mono-to-binaural conversion with intrinsic learning of Ambisonics-like channels. We can summarise the main contributions into the following points:

- We propose a novel approach for mono-to-binaural music generation, inspired from the Ambisonics format. This method works by predicting shared time-frequency internal representation alongside gain parameters. We use these to construct the binaural representation from its mono counterpart, taking inspiration from the Ambisonic format for the representation and decoding to obtain the final result;

- We propose a new hierarchical spatio-visual module for conditioning binaural audio generation. This conditioning is used in the latent space to obtain crude representations that are decoded into the channels mentioned above;

- To the best of our knowledge, our work is the first to incorporate both multi-scale and multi-modality visual prior knowledge with learnable position encoding, to obtain a full representation of the observed surroundings – a key component for achieving the spatiality of binaural audio;

- We demonstrate the efficacy of our approach on three commonly used binaural audio-visual music datasets. The proposed method obtains state-of-the-art results, cementing our hypothesis on combining traditional and deep-learning methods for more robust mono-to-binaural. Moreover, we extended our work to out-of-domain, general audio with a plug-in-play HiVi module adaptation.

The rest of the manuscript is organised as follows: Section 2 briefly describes the previous works in this domain, Section 3 introduces our proposed solution, Section 4 validates our approach, while Section 5 provides general conclusions.

## 2 BACKGROUND & RELATED WORKS

### 2.1 CONDITIONAL AUDIO GENERATION

Conditional audio generation has advanced significantly in recent years, largely propelled by breakthroughs in conditional image modelling. This progress spans a wide range of domains, from specialised tasks such as speech synthesis (Lee et al., 2025; Wang et al., 2025) and music generation (Mariani et al., 2024), to more general approaches involving multimodal conditioning (Tian et al., 2025). Building on these developments, recent works have proposed systematic taxonomies of conditional audio generation, typically distinguishing between tasks such as text-to-audio, image-to-audio, and joint audio–visual generation (Hayakawa et al., 2025). Our work focuses on generating a binaural audio from its mono counterpart, conditioned on visual cues.

**Text-to-Audio generation:** Early works for this task are closely linked to TTS systems. However, this task has been recently extended to open-domain audio generation with the introduction of AudioLDM (Liu et al., 2023), a text-guided latent diffusion model which operates in the latent space of a spectrogram-based VAE, aligning the captioning with the provided audio during training. Follow-up works, such as AudioLDM2 (Liu et al., 2024a) or Tango2 (Majumder et al., 2024), have focused on generating higher quality audio or adhering to user preference, optimising the listening experience. Due to limited data, text-based approaches for audio generation do not take into account sound direction, solely measuring the prompt alignment using contrastive models (Wu et al., 2023).

**Vision-conditioned audio**: These models leverage pretrained visual encoders with audio generation backbones, enabling image-to-audio or video-to-audio generation. One common approach (Wang et al., 2024) is to make use of lightweight mappers to connect vision foundation models to audio generators without fully re-training, while others have drawn inspiration from LLM training strategies

to introduce token-based audio generation (Mehta et al., 2025). This approach has sparked many research directions, showcasing the need for low-bit but precise neural audio encoders (Ji et al., 2025) and for shared, modality-independent, embedding spaces (Girdhar et al., 2023). However, both of these directions are still mainly operating in the single-channel audio domain, while our work extends not only to general-purpose stereo but on binaural audio.

**Spatial audio generation:** Recently, generative models have expanded to address the more complex problem of spatial audio generation. Recently, (Kim et al., 2025) proposed ViSAGe, a silent video-to-spatial audio generation method, which uses First-Order Ambisonics (FOA) (see Section 2.3 for details) extracted from silent Field-of-View (FoV) videos. Moreover, they introduce a new dataset, YT-Ambigen, featuring in-the-wild videos with spatial audio. Another direction is represented by OmniAudio (Liu et al., 2025), which introduces the 360° panoramic view to spatial audio generation, since FoV inputs do not capture the full spatial context. To train their method, the authors introduce the Sphere360 dataset, containing 360° videos and their associated FOAs. While the aforementioned systems rely on visual conditioning, (Sun et al., 2025) proposed SpatialSonic, a language-driven spatial synthesis model. Their generation method can be conditioned on textual descriptions as well as a variety of types of visual inputs, such as bounding boxes, or interactive actions, *i.e.* selecting which object is the sounding one. However, a major drawback of these methods is represented by the lack of fine-grained control over the output, *e.g.* sounds are generated based on the extracted semantics of the provided input, which is not suitable for music, where small inconsistencies are extremely noticeable. One alternative, INRAS (Su et al., 2022), aims at generating impulse responses (IRs), w.r.t. the room configuration and the sounding and listening locations, to be convolved with the mono audio to obtain the spatial variant. This approach is completely different from ours, in which we generate the Ambisonics channels directly instead of relying on additional signals to serve as filters.

## 2.2 Mono-to-binaural using visual information

Most prior works have treated music generation as a monophonic task, producing signals with a single channel. In practice, however, the majority of real-world audio is stereophonic, thus spatial , reflecting both the binaural nature of human hearing and the widespread use of headphones in everyday listening. To bridge the gap between mono and stereo audio, several studies have incorporated visual information to guide the spatial positioning of sounds, thereby improving object localisation and enhancing the immersive quality of the generated audio. Pioneering this domain, (Gao & Grauman, 2019) proposes the combination of a spectrogram-based UNet for binaural generation. Inside the UNet bottleneck, the visual features extracted by a ResNet-18 model pretrained on ImageNet are concatenated, which has become outdated. Our solution is to create an ensemble of methods that extract multimodal visual information, guaranteeing a more robust solution.

Sep-stereo (Zhou et al., 2020) aims to improve stereophonic learning by also including audio-visual source separation. By allowing parallel training on mono audio separation aided by visual information, they improve the stereo generation in the context of scarce binaural music data. This strategy has also been applied more recently by CLUP (Li et al., 2024), combined with a diffusion strategy. PseudoBinaural (Xu et al., 2021), as the name implies, focuses on generating binaural data without mono-stereo pairs. Using visual-coordinate mapping, their focus is on producing Ambisonics coefficients and HRIR filters from spatial priors, which can be applied to the mono signals to encode their location inside a stereo audio. Beyond Mono2Binaural (Beyond M2B) (Parida et al., 2022) is the first work to add a Depth network to improve the results of previous works, with a decoder that attends to both image-audio and depth-audio features. SAGM (Li et al., 2023) uses a GAN-style method for generating music, with a discriminator to decide between features of real binaurals and generated ones, which are concatenated with video features.

Recently, CMC (Liu et al., 2024b) proposed a dual-encoder approach for the left and right channels, alongside a new cross-matching loss. Finally, CCStereo (Chen et al., 2025) makes better use of the temporal dimension in both audio and video data with the introduction of a conditional normalisation layer and audio-video alignment. As such, previous methods tend to focus on better separation of the sounding elements or channels, to obtain better alignment, while under-exploring available prior information in both domains. In our work, we introduce several novelties in both the internal processing of our proposed solution, which captures the audio (mono) prior, as well as focusing on extracting more relevant information using multimodal vision approaches and hierarchically extracted features.

## 2.3 AMBISONICS NOMENCLATURE AND CODING

Ambisonics is a spatial audio technique for representing the sound field description around a listening point using spherical harmonic decomposition. Instead of directly capturing the signal that should be played on speakers placed at certain locations, Ambisonics encodes the sound itself, allowing for arbitrary decoding for any speaker layout. Let $s(t) \in \mathbb{R}$ be the value of an audio waveform at timestep $t$ of a single sounding source $s$, *e.g.* a voice, an instrument or a noise, and $(r, \theta, \phi)$ be the source polar coordinates. To encode this, a special *Ambisonics Channel Signal* (ACN) is used:

$$\text{ACN}_m^{(l)}(t) = s(t) S_m^{(l)}(\theta, \phi) , \tag{1}$$

where $S_m^{(\ell)}$ is the spherical harmonic function, $\ell \geq 0$ denotes the Ambisonics order, and $m$ denotes the Ambisonics index, while respecting the $-l \leq m \leq l$ constraint.

In practice, only the first-order Ambisonics ($\ell = 1$, abbreviated as FOA) are frequently used, where the following channels are defined using truncated spherical harmonic expansions, representing dipoles for each Cartesian axis: **W channel**: omnidirectional components (zero order), which capture the sound from all directions equally, similar to the mono audio format; **X channel**: contains differences on the front-back axis, giving the audio more depth and **Y channel**: left-right pattern, used for giving the directional feeling of audio. For a full 3D experience, the **Z channel** is also used to allow for up-down direction. These channels are computed as follows, using the initial source $s(t)$ and its spherical positions $(\theta, \phi)$:

$$\text{ACN}_0^{(0)} = W(t) = s(t), \tag{2}$$

$$\text{ACN}_1^{(1)} = X(t) = s(t) \cos\theta \cos\phi, \tag{3}$$

$$\text{ACN}_1^{(-1)} = Y(t) = s(t) \sin\theta \cos\phi, \tag{4}$$

$$\text{ACN}_1^{(0)} = Z(t) = s(t) \sin\phi. \tag{5}$$

Since an audio recording can contain multiple sounding objects, each with its own spatial position, obtaining the final audio is done by summing up all the representations. Considering the classical stereo position, where left $L(t)$ and right $R(t)$ speakers are positioned at ground level, *i.e.* $\phi = 0$, and opposite angles, *i.e.* $\alpha = \theta_L = -\theta_R$, the FOA for $N$ sources must also account for the position of the speakers playing each sound, individually:

$$\text{L(t)} = \sum_{i=1}^{N} \frac{1}{\sqrt{2}} W_i(t) + X_i(t) \cos\alpha_i + Y_i(t) \sin\alpha_i, \tag{6}$$

$$\text{R(t)} = \sum_{i=1}^{N} \frac{1}{\sqrt{2}} W_i(t) + X_i(t) \cos\alpha_i - Y_i(t) \sin\alpha_i. \tag{7}$$

This formulation highlights how Ambisonics provides a structured intermediate representation for spatial audio, which we leverage as the foundation of our approach. Moreover, using this approach also incorporates the panning of the speakers w.r.t. the source's original position.

## 3 APPROACH: HIVIBIX

The overall architecture of the proposed model is depicted in Fig. 1. We denote the input data with $v \in \mathbb{R}^{(T_v \times C_v \times H_v \times W_v)}$ for the video sequence, where $T_v$, $C_v$, $H_v$ and $W_v$ denote the frame, channel, height and width dimension, respectively. Let $a_{\text{bi}} \in \mathbb{R}^{(C_a \times T_a)}$ be the binaural audio, with $C_a$ and $T_a$ denoting the channels and sample (time) sizes. From $a_{\text{bi}}$ we extract the mono signal $a_{\text{m}}$ by summing the channels and obtain the $S_m \in \mathbb{C}^{(1 \times F \times T)}$ spectrogram by applying the Short-Time Fourier Transform (STFT) operation, with $F$ and $T$ denoting the frequency and the time bins, respectively. Since we are using the mono representation as the $W$ channel from the Ambisonics format, we use the same notation throughout the text, *i.e.* $W = S_{\text{m}}$. The goal of our network is to generate the other Ambisonics channels, *i.e.* $\hat{X}$ and $\hat{Y}$ and to combine them into the final binaural spectrogram $\hat{S}_{\text{bi}} \in \mathbb{C}^{(2 \times F \times T)}$ to obtain the final binaural audio $\hat{a}_{\text{bi}} \approx a_{\text{bi}}$. To implement this pipeline, we employ several modules: an Encoder-Decoder strategy, presented in Section 3.1, with a latent space conditioned on a new visual encoder, detailed in Section 3.2, followed by the Ambisonics FiLM layer, which ensures our internal representation, described in Section 3.3.

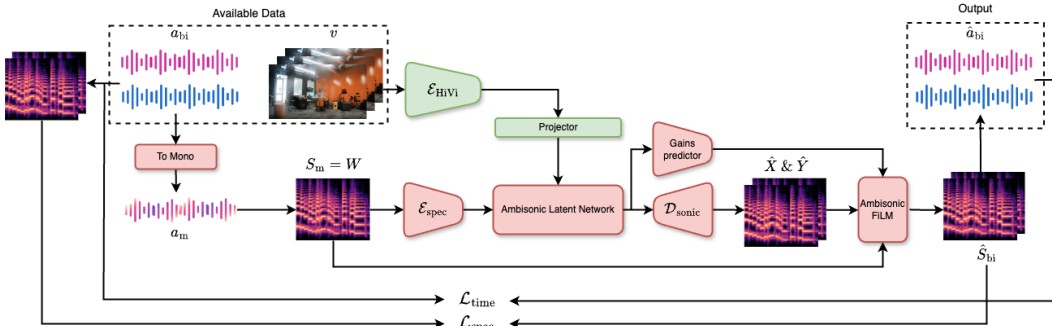

Figure 1: Proposed training scheme. We start from a video and its associated binaural, that is converted to mono and further to its spectrogram representation. Next, we encode it into a latent space that is conditioned using our new hierarchical vision (HiVi) encoder. We then decode this into both the Ambisonics channels and required gains, which are used to produce the binaural spectrogram and the final audio. We use two classes of losses: in the waveform and spectrogram domains.

## 3.1 MONO-TO-AMBISONICS ENCODER & DECODER

AudioLDM (Liu et al., 2023) has recently emerged as a powerful framework capable of learning intricate audio input-output relationships in the time-frequency domain, based on textual conditioning. Since our goal involves mapping mono audio to its binaural representation with the aid of visual cues, this model provides a natural and suitable inspiration for our overall architecture design. First, we start by designing a Convolutional encoder $\mathcal{E}_{\mathrm{spec}}$ that reduces the input spectrogram into a compact, more abstract representation. The Decoder $\mathcal{D}_{\mathrm{sonic}}$ mirrors this structure, and generates the output corresponding to the desired channels. Because the input-output representations are not semantically similar *i.e.* encoding mono channels but decoding difference-related channels $(X, Y)$, we omit the skip connections between $\mathcal{E}_{\mathrm{spec}}$ and $\mathcal{D}_{\mathrm{sonic}}$. We construct the latent space by employing a conditional residual Attention UNet (ResAttnUNet) for a guided transformation of the mono latent representation towards the Ambisonics one, using the conditioning vector extracted by $\mathcal{E}_{\mathrm{HiVi}}$. For both the magnitude and the phase we use the complex representation of $S_m$ and individual networks to predict the $\hat{X}$ and $\hat{Y}$ channels, as opposed to a multi-channel output network, to allow each Encoder-Decoder structure to focus on channel-specific features. Moreover, we convert the Ambisonics position and gain coefficients into learnable parameters, enabling the model to have full control over the desired distribution. As in the case of channels, we treat these parameters independently for each modality, *i.e.* magnitude and phase. For a more detailed view about the implementation of this module, check Appendix A.

## 3.2 HIERARCHICAL VISUAL (HIVI) ENCODER

To improve the audio, especially music, generation, we added a new module for extracting conditions from the provided video, as depicted in Fig. 2. The binaural audio benefits from visual cues, as each sounding object can be associated with its position from the video, information which we inject in the Ambisonics latent space. To obtain this conditioning vector, firstly, we are selecting an anchor image $v^{(i)} \in \mathbb{R}^{(C_v \times H_v \times W_v)}$ to extract prior knowledge that would guide the generation. From the anchor image, we extract a list of sounding objects from the image using YOLOv8 [1] by selecting the bounding boxes associated with the `person` label. We chose this approach as most instruments require a human operator, and several solutions are unable to detect all the instruments from the used datasets. We used YOLO as a faster and lighter alternative to other object detection approaches. We crop the regions using the predicted bounding box, enlarged by 20%, and treat them as local information, in contrast to the full image, which serves as global context. The extracted knowledge domains are three-folded: scene, position and depth. For general and depth features, we employ a cross-attention structure to combine them into a single representation for their corresponding modality.

---

[1]YOLOv8 available at: `https://github.com/ultralytics/ultralytics`

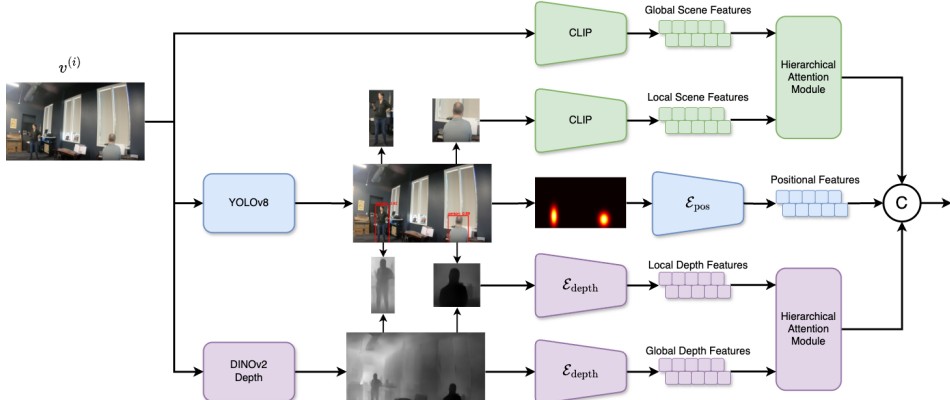

Figure 2: Hierarchical Visual encoder. We extract both local and global features from multiple modalities, *i.e.* RGB, depth and position, computed solely based on the provided image.

We use the image encoder from CLIP (Radford et al., 2021) to extract both local and global scene features. The local features to distinguish between different instruments, while the global ones are responsible for giving the overall audio dynamics of the scene (indoors, sound reflecting elements such as walls or other obstacles, microphone positioning, *etc.*). For positioning, we select 2D Fourier Features for Positional Encoding (FFPE) (Li et al., 2021) to generate prior knowledge about the location of the sounding objects in the frame, Fig. 2 (middle row). We chose this method for representing this knowledge because a video frame can contain multiple audio sources. By selecting the Gaussian kernel provided by FFPE, and summing them into a single position image $P \in \mathbb{R}^{(1 \times H_v \times W_v)}$, we are able to distinguish between both separate and overlapping sounding objects. Finally, depth information is also useful, especially for the $X$ channel. As such, we use a depth estimation model based on DINOv2 (Oquab et al., 2024). As in the scene feature case, we extract both global and local features. Global depth features are useful for better understanding the room configuration, which can help for internal modelling of reverberations, while local ones provide the necessary distance comprehension to each sounding object.

Combining the global and local prior knowledge is an essential step to obtain an efficient conditioning. As such, we carefully craft the Hierarchical Attention Module, which is an adaptation of the classical cross-attention. This block receives the global features as queries, while the keys and values are the local ones. The global representation selectively gathers fine details from the local parts, allowing them to be informative inside the broader context, for each modality that has this component. Finally, we concatenate all the modalities to obtain a single conditioning vector, which is further projected inside the latent Ambisonics space.

### 3.3 AMBISONICS FiLM

Our method closely follows the Ambisonics format for binaural generation of music and general audio. Since most datasets do not contain the Ambisonics representation, and only the final stereo is available, we based our experiments on internally learning the FOA-like spectrograms. This is facilitated by the new Ambisonics FiLM layer, also described in Algorithm 1. As per previous steps, we compute the magnitude and phase independently. Moreover, we use the mono (unidirectional) signal as a replacement for the $W$ channel in both cases. For phase reconstruction, we predict only the interaural phase difference (IPD), as other works suggested (Pan et al., 2021). The main idea of the Ambisonics FiLM layer is to use the available prior, *i.e.* $W$, and predicted, *i.e.* $\hat{X}$ and $\hat{Y}$ channels, and to force the reconstruction in the same way as the Ambisonics decoding of binaural audios, as shown in Section 2.3.

---

**Algorithm 1:** Ambisonics FiLM Layer

---

**Data:** Mono spectrogram magnitude and phase: $W = \{W_M, W_P\} \in \mathbb{R}^{(1 \times F \times T)}$;

Predicted Ambisonics channels for magnitudes and phases: $\{\hat{X}, \hat{Y}\}_{\{M,P\}} \in \mathbb{R}^{(1 \times F \times T)}$;

Predicted position and gain coefficients: $\hat{\alpha}_{\{M,P\}}^{\{X,Y\}}, \hat{\beta}_{\{M,P\}} \in \mathbb{R}$

**Result:** Predicted binaural spectrogram: $\hat{S}_{\text{bi}} \in \mathbb{C}^{(2 \times F \times T)}$

$$\widehat{M}_{\text{bi}}^L = \left( W_M + \cos(\hat{\alpha}_M^X)\,\hat{X}_M + \sin(\hat{\alpha}_M^Y)\,\hat{Y}_M \right) \beta_M^{-1}$$

$$\widehat{M}_{\text{bi}}^R = \left( W_M + \cos(\hat{\alpha}_M^X)\,\hat{X}_M - \sin(\hat{\alpha}_M^Y)\,\hat{Y}_M \right) \beta_M^{-1}$$

$$\widehat{\text{IPD}} = \left( W_P + \cos(\hat{\alpha}_P^X)\,\hat{X}_P - \sin(\hat{\alpha}_P^Y)\,\hat{Y}_P \right) \beta_P^{-1}$$

$$\widehat{S}_{\text{bi}} = \left[\, \widehat{M}_{\text{bi}}^L,\ \widehat{M}_{\text{bi}}^R \,\right] \cdot \exp\!\left( j \left[ W_P + \widehat{\text{IPD}},\ W_P - \widehat{\text{IPD}} \right] \right)$$

---

## 4 EXPERIMENTS

### 4.1 IMPLEMENTATION DETAILS

For training, we used three distinct datasets: FAIR-Play (Gao & Grauman, 2019), Music-Stereo (Xu et al., 2021) and YT-Music (Morgado et al., 2018). More details about the datasets can be found in Appendix B. We used a sample rate of 16kHz, following recommendations from previous works. The generated spectrograms were computed with a rectangular window of 1024 samples, which also defined the number of FFT points, and a hop size of 25% (256 samples). These settings yielded spectrograms of size $513 \times 512$, corresponding to approximately 8.2 seconds of audio. We removed the last frequency bin and applied zero-padding during iSTFT reconstruction. No data augmentation was employed. The models were optimized using a compositional loss that jointly accounted magnitude, phase and time domain changes, as detailed in Appendix C. Training was performed from scratch for up to 500 epochs with the AdamW optimizer (Loshchilov & Hutter, 2019), a decaying learning rate scheduler with 5 epochs for patience to avoid plateaus, and an early stopping mechanism triggered if the validation loss did not improve within 15 epochs. All experiments were performed on an NVIDIA A100 with 40GB of VRAM, using minibatches of 16 examples. For testing, we used the same parameters as (Chen et al., 2025) to obtain comparable results in the frequency domain. Each signal was split into two parts and zero-padded at both ends. After prediction, we reconstructed the original, full 10 seconds samples by combining only the relevant parts of each segment.

### 4.2 QUANTITATIVE RESULTS

We compared our approach against prior works that employed visual conditioning for mono to binaural audio conversion. To evaluate the performance, we adopted metrics spanning both the time and spectrogram domains. Specifically, we used the STFT $L2$ distance (STFT) to measure differences in the time-frequency domain, and the envelope distance (ENV) served as a well established metric in the time domain. Moreover, we assessed the overall quality of the generated binaural signals, using the signal-to-noise ration (SNR). Further implementation details and metrics are provided in the Supplementary Materials.

All baseline results were either taken from the respective papers or obtained from publicly available re-implementations, where possible. Table 1 presents the results on the FAIR-Play dataset 10-splits and 5-splits, respectively, while Table 2 is designated for the results on Music-Stereo and YT-Music.

Compared to other methods, our approach achieves superior performance across all datasets. Firstly, the results on FAIR-Play 10-splits indicate that the model is capable of rendering room acoustics across different angles and instruments. The results on the new 5-split (Xu et al., 2021), which reduces the overlap between the splits, illustrate that the model is also able to generalize to unseen cases. On the YouTube datasets, which include a multitude of instruments and other noises, we can observe that our model is capable of understanding the interdependency between sounding sources, their position and other sounding objects or people. Importantly, the performances on Music-Stereo and YT-MUSIC, datasets containing both indoor and outdoor scenes, highlight the ability of our model to generalize beyond the constrained setting of enclosed rooms present in the FAIR-Play dataset.

| Methods | FAIR-Play 10 splits | | | FAIR-Play 5 splits | | |
|---|---|---|---|---|---|---|
| | STFT ↓ | ENV ↓ | SNR ↑ | STFT ↓ | ENV ↓ | SNR ↑ |
| Baseline (Mono-as-Stereo) | 2.356 | 0.281 | 3.565 | 1.828 | 0.240 | 0.000 |
| Mono2Binaural *[CVPR '19]* | 0.836 | 0.132 | - | 1.024 | 0.145 | 4.968 |
| Sep-stereo *[ECCV '20]* | 0.879 | 0.135 | 6.422 | 0.906 | 0.136 | 5.221 |
| PseudoBinaural *[CVPR '21]* | 0.878 | 0.134 | 5.316 | 0.944 | 0.139 | 5.124 |
| Beyond M2B *[WACV '22]* | 0.909 | 0.139 | 6.397 | 0.909 | 0.139 | 6.397 |
| SAGM *[KBS '23]* | 0.851 | 0.134 | 7.044 | - | - | - |
| CMC *[ICASSP '24]* | 0.849 | 0.133 | - | 0.912 | 0.141 | 6.238 |
| CLUP *[CVPR '24]* | 0.787 | 0.128 | 7.546 | - | - | - |
| CCStereo *[ACM MM '25]* | 0.823 | 0.132 | 7.144 | 0.883 | 0.137 | 6.475 |
| **HiViBi** (Ours) | **0.6319** | **0.123** | **7.629** | **0.880** | **0.126** | **6.483** |

Table 1: Comparison of HiViBi against other methods on FAIR-Play dataset, on both splits. The best results are bolded, while the second-best are underlined.

| Methods | Music-Stereo | | | YT-Music | | |
|---|---|---|---|---|---|---|
| | STFT ↓ | ENV ↓ | SNR ↑ | STFT ↓ | ENV ↓ | SNR ↑ |
| Baseline (Mono-as-Stereo) | 3.400 | 0.369 | 0.000 | 1.067 | 0.180 | 0.000 |
| Mono2Binaural *[CVPR '19]* | 0.942 | 0.138 | 8.255 | 0.501 | 0.110 | 6.712 |
| Sep-stereo *[ECCV '20]* | - | - | - | 1.051 | 0.145 | 4.779 |
| PseudoBinaural *[CVPR '21]* | 0.891 | 0.132 | 8.419 | 0.489 | 0.109 | 7.601 |
| Beyond M2B *[WACV '22]* | 0.670 | 0.108 | 10.754 | 1.070 | 0.148 | 4.542 |
| SAGM *[KBS '23]* | 0.875 | 0.126 | 5.601 | 0.875 | 0.126 | 5.601 |
| CMC *[ICASSP '24]* | 0.759 | 0.113 | - | - | - | - |
| CLUP *[CVPR '24]* | - | - | - | 0.856 | 0.124 | 5.711 |
| CCStereo *[ACM MM '25]* | 0.624 | 0.097 | 12.985 | 0.432 | 0.102 | **8.245** |
| **HiViBi** (Ours) | **0.331** | **0.070** | **14.363** | **0.260** | **0.073** | 7.805 |

Table 2: Comparison of HiViBi against other methods on Music-Stereo and YT-Music datasets. The best results are bolded, while the second-best are underlined.

### 4.3 QUALITATIVE RESULTS

We further demonstrate the quality of our proposed solution through both time and frequency domain evaluations on the FAIR-Play dataset. Results for the remaining datasets are included in the Supplementary Materials. Fig. 3 illustrates the ground truth and predicted binaural signals in the time domain. We averaged them over 40 samples, for better visualisation, creating piecewise-like representations. Our predictions closely follow the real ones, in multiple scenarios: on the first row, where there is only one instrument close to the left of the microphone and reflective element (the door) close to the right of it, capturing the delayed sounds; on the second row, where the trumpet is further back but leans towards a side; and on the final row where two instruments are in completely opposite locations.

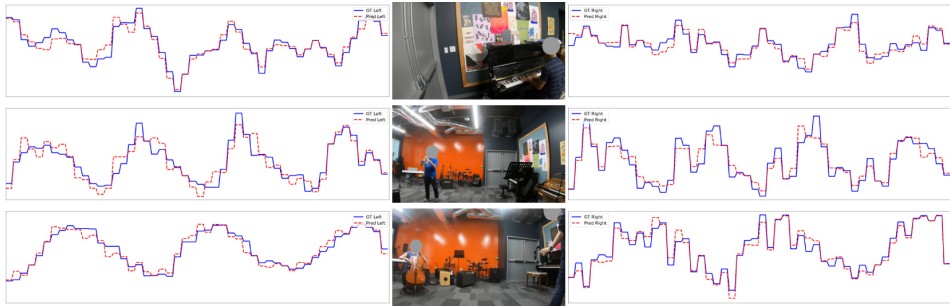

Figure 3: Piecewise signals, ground truth (in blue) and predicted (in red) for the left and right channels. Each row represents an individual example. The image in the middle showcases the localisation of the sounding objects for easier discrimination in the audio.

A similar phenomenon can be observed in the time-frequency domain, as illustrated in Fig. 4. Our predictions respect a similar pattern to the original binaural spectrograms. The first example showcases a smothering effect, especially in the higher frequencies. This is similar to a low pass filter, indicating that our network does not fully capture the finest details. The last example shows the opposite effect, where small gapes in the original spectrogram appear as larger in the prediction, indicating that low-energy regions tend to be more persistent in our predictions than in the case of real binaural.

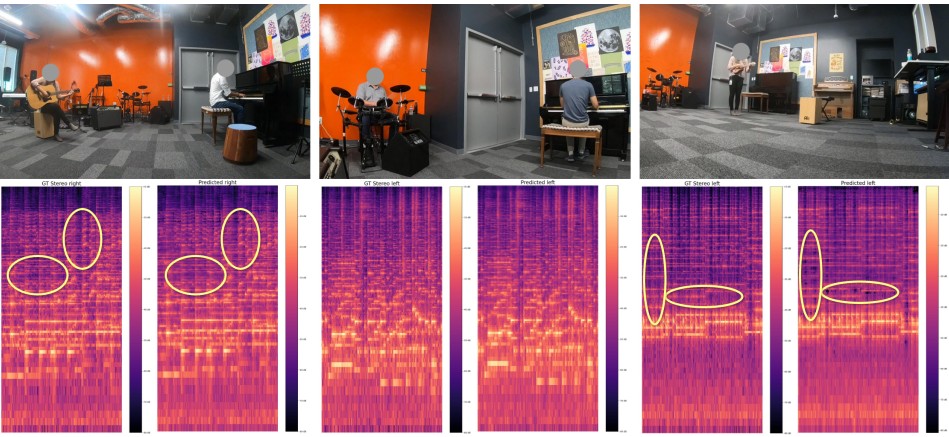

Figure 4: Spectrogram comparison. The first row depicts the visual image, while the second row showcases the ground truth and predicted spectrogram. We selected only one channel for each example, for easier visualisation.

### 4.4 ABLATION STUDIES

To assess the contribution of each component in our approach, we conducted an ablation study, with the results summarized in Table 3. For consistency, all ablation experiments were trained only on the first split of the FAIR-Play (10-split) dataset. We systematically evaluate the following aspects of our design: (i) the output representation, where we compare our Ambisonics-like format against direct prediction of magnitude and phase spectrograms; (ii) the loss function, contrasting our proposed compound loss $\mathcal{L}_C$ with a standard end-to-end $L_2$ loss on the predicted and ground-truth binaural signals; and (iii) the design of the visual encoder, where we separately evaluate the contributions of CLIP embeddings, depth cues, and positional information. The HiVi column indicates the use of our hierarchical visual encoder, which integrates both global and local features, as opposed to global features alone.

| Components | | | | | | STFT ↓ | ENV ↓ | SNR ↑ |
|---|---|---|---|---|---|---|---|---|
| AS | $\mathcal{L}_C$ | CLIP | Depth | Pos | HiVi | | | |
| ✗ | ✓ | ✓ | ✓ | ✓ | ✓ | 1.492 | 0.199 | 4.644 |
| ✓ | ✗ | ✓ | ✓ | ✓ | ✓ | 0.715 | 0.132 | 6.938 |
| ✓ | ✓ | ✗ | ✓ | ✓ | ✓ | 0.692 | 0.128 | 7.165 |
| ✓ | ✓ | ✓ | ✗ | ✓ | ✓ | 0.700 | 0.129 | 6.912 |
| ✓ | ✓ | ✓ | ✓ | ✗ | ✓ | 0.712 | 0.132 | 6.868 |
| ✓ | ✓ | ✓ | ✓ | ✓ | ✗ | 0.734 | 0.134 | 6.633 |
| ✓ | ✓ | ✓ | ✓ | ✓ | ✓ | 0.669 | 0.125 | 7.489 |

Table 3: Ablation study showing performance impact of individual components of our proposed method. ✗ denotes a removed component while ✓ indicates a kept one.

Our findings reveal two key insights. First, Ambisonics-like representations, multimodal conditioning, and hierarchical visual encoding each provide substantial and complementary performance gains. Second, while each component individually achieves results comparable to the state of the art, only their combination establishes a new performance benchmark. These results highlight two

promising research directions: improving the intermediate representation of audio and enhancing the extraction of conditioning signals from other visual modalities.

## 5 DISCUSSIONS & CONCLUSION

### 5.1 LIMITATIONS

Our approach relies mainly on two components: the Ambisonics and Hierarchical Visual, each with its own potential limitation. Firstly, the Ambisonics-like format is only enforced by the Ambisonics FiLM layer, described in Section 3.3, due to lack of real Ambisonics data to be compared with. However, since YT-Music dataset contains the Ambisonics audios, we present a comparison between the intermediate feature maps and real Ambisonics in Appendix E. We show that using this novel layer results in the learning of Ambisonics-like channels without any direct Ambisonics supervision. For the vision component, the fine-grained features are extracted based on the object detection model prediction, which can be incorrect or ambiguous in some cases, *e.g.* when there are several people in the frame. Moreover, using the person class for instrument detection can be disadvantageous, especially when considering other sounding objects, *e.g.* speakers. To mitigate this, we also kept the global features branch. Finally, rapidly changing videos can also induce problems in our framework, as illustrated in Appendix G . We took advantage of the dataset distribution, which poses mostly static videos, and relied on single-frame predictions. However, in real scenarios, aggregation of multiple, different frames might be a more well suited approach for dynamic videos.

### 5.2 CONCLUSION

In this paper, we presented HiViBiX, a mono-to-binaural generation framework that leverages multimodal visual priors from a single image to transform mono audio into its corresponding binaural counterpart. The key innovation lies in internally predicting Ambisonics-like channels and gains, which serve as an intermediate representation for decoding the binaural signal. Our method overcomes the limitations of prior works by exploiting the mono input, and by enriching the visual conditioning beyond global contrastive features. To achieve this, we also incorporate depth and position information from both local and global cues, arranging them in a hierarchical manner. Extensive experiments across multiple datasets demonstrated that HiViBiX consistently outperforms existing methods and sets a new state of the art in the domain.
This works paves the way for future research in the binaural generation domain, by providing the necessary introduction to new representation formats, as well as a method for combining multivision information.

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

## A  MONO-TO-AMBISONICS ENCODER-DECODER DETAILS

Our networks operate in the time-frequency domain, on two separate branches: magnitude and phase. Firstly, to ease the learning process, we adopted the logarithmic scale for magnitude and normalised the phase to the $\pm 1$ range. The two branches are identical, and consist of the spectrogram encoder $\mathcal{E}_{\text{spec}}$, Ambisonics Latent Network (ALN) and Ambisonics-like decoder $\mathcal{D}_{\text{sonic}}$. Since the information contained within the spectrograms is spatially correlated (both in time and frequency axis), we use the $\mathcal{E}_{\text{spec}}$ to reduce the dimensionality of the input from $512 \times 512 \rightsquigarrow 128 \times 128$, effectively reducing the computational cost of the ALN by 4. To do so, we employ a 3-layered CNN, with ReLU activations and max pooling. For ALN, we employ a 4-layer ResAttnUNet, with cross-attention between the input and the projected visual encodings. For projection, we use a fully-connected layer to reduce the dimensionality, in order to match the input size. Finally, the prediction is then upscaled using transposed convolutions inside $\mathcal{D}_{\text{sonic}}$.

We discovered empirically that employing this framework for each Ambisonics channel, *i.e.* $\hat{X}$ and $\hat{Y}$, works better than predicting them as two separate channels of the same network. Fig. 5 describes this internal process visually. For the gains and panning coefficients, we treat them as learnable parameters for the Ambisonics FiLM channels to use.

## B  DATASETS

**FAIR-Play:** Firstly, we used the most popular dataset for this task, the FAIR-Play dataset. We followed both the initial (10-split) (Gao & Grauman, 2019) and newly organised (5-split) (Xu et al., 2021) set, with the latter being created to better showcase the generalisation capabilities of models trained on this dataset. It consists of 1,871 10-second clips, recorded in a music room, from different angles, each clip being accompanied by its binaural version.

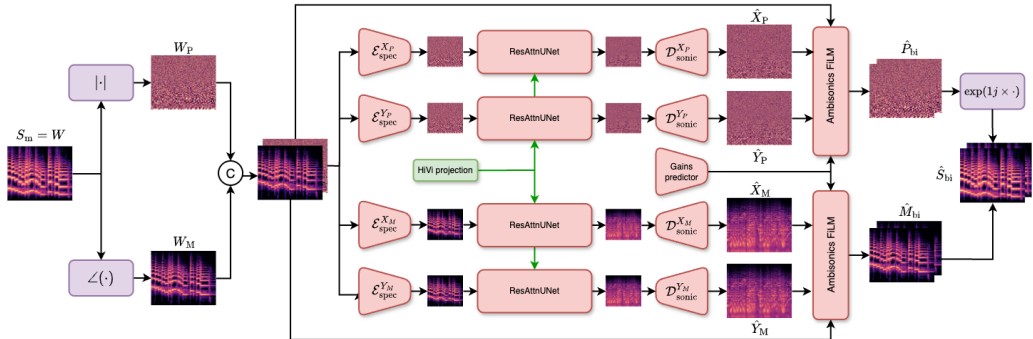

Figure 5: Detailed view of our proposed network. Top branches are dedicated to the phase processing, while bottom ones are for magnitude approximation. We omit the normalisation steps for better clarity. The HiVi projection (in green) is shared in all branches.

**Music-Stereo:** Introduced in (Xu et al., 2021), this dataset is composed of music audio clips recorded from 21 different instruments in solo and duet parts. Initially containing 1,120 videos (from which, at time of writing, only 1,145 are still available), it is the largest available binaural dataset, totalling almost 50 hours, $10\times$ larger than FAIR-Play. We followed the same pre-processing steps as the initial authors to obtain 20,096 10-second clips, from which we used aan80/10/10 split for training/validation/testing.

**YT-Music:** This dataset is comprised of 397 (from which only 358 are still available) YouTube videos in the 360° format and audio in the Ambisonics format. We converted each audio to a binaural one using the same split and processing steps as (Gao & Grauman, 2019). This is the most challenging dataset, as it contains a greater variation in overall scenes and present sources. We selected only the clips that contain at least one human and at most 20, resulting in 10,477 clips.

## C LOSSES & METRICS DETAILS

Following the notations from Section 3, we can define the following losses for the time and time-frequency domains. Firstly, since our network computes the predicted magnitude and phase of the binaural signal, we are using modified versions of the $L2$ magnitude and angle loss as our losses for spectrograms. These are presented in Eqs. (8) and (9), respectively.

$$\mathcal{L}_{\text{MAG}}\left(S_{\text{bi}}; \hat{S}_{\text{bi}}\right) = \left(|S_{\text{bi}}^L| - |\hat{S}_{\text{bi}}^L|\right)^2 + \left(|S_{\text{bi}}^R| - |\hat{S}_{\text{bi}}^R|\right)^2, \tag{8}$$

where $|\cdot|$ denotes the modulus operator.

$$\mathcal{L}_{\text{IPD}}\left(S_{\text{bi}}; \hat{S}_{\text{bi}}\right) = |\angle(S_{\text{bi}}^L - S_{\text{bi}}^R) - \angle(\hat{S}_{\text{bi}}^L - \hat{S}_{\text{bi}}^R)|, \tag{9}$$

where $\angle(\cdot)$ denotes the phase angle of the complex spectrogram.

For the time domain, however, we are relying on two loss adaptations: waveform distance, depicted in Eq. (10), and the signal-to-noise ratio (SNR), see Eq. (11). Our experiments show that including an end-to-end loss, such as WAV, helps in better regularising this domain-specific caveats, such as offering more consistency between the sample transitions.

$$\mathcal{L}_{\text{WAV}}(a_{\text{bi}}; \hat{a}_{\text{bi}}) = \frac{1}{T_a} \sum_{c=1}^{C_a} \sum_{t=1}^{T_a} \left(a_{\text{bi}}^{(c,t)} - \hat{a}_{\text{bi}}^{(c,t)}\right)^2, \tag{10}$$

$$\mathcal{L}_{\text{SNR}}(a_{\text{bi}}; \hat{a}_{\text{bi}}) = \Gamma - \frac{\mathbb{E}(a_{\text{bi}})}{\mathbb{E}(a_{\text{bi}} - \hat{a}_{\text{bi}})}, \tag{11}$$

where $\mathbb{E}(\cdot)$ denotes the expectation operator and $\Gamma$ is an empirically upper bound selected w.r.t the used dataset. For Music-Stereo, we used $\Gamma = 20$, and for the rest of the experiments, we set $\Gamma = 15$.

To accommodate the different values of our losses, we used a compound loss, with the weights

$$\mathcal{L}_C = \mathcal{L}_{\text{MAG}} + \mathcal{L}_{\text{IPD}} + 100\mathcal{L}_{\text{WAV}} + 0.1\mathcal{L}_{\text{SNR}}. \tag{12}$$

For metrics, we focus on comprehensive ones, such as the Short-Time Fourier Transform (STFT) distance, see Eq. (13) for the time-frequency domain, and signal envelope (ENV), Eq. (14) for the time domain. Additionally, we report the SNR, Eq. (15), to quantify the quality of our generated binaurals.

$$\text{STFT}\left(S_{\text{bi}}; \hat{S}_{\text{bi}}\right) = \|S_{\text{bi}}^L - \hat{S}_{\text{bi}}^L\|_2 + \|S_{\text{bi}}^R - \hat{S}_{\text{bi}}^R\|_2, \tag{13}$$

where $\|\cdot\|_2$ denote the Euclidean distance.

$$\text{ENV}(a_{\text{bi}}; \hat{a}_{\text{bi}}) = \|E[a_{\text{bi}}^L] - E[\hat{a}_{\text{bi}}^L]\|_2 + \|E[a_{\text{bi}}^R] - E[\hat{a}_{\text{bi}}^R]\|_2, \tag{14}$$

where $E[\cdot]$ denotes the envelope of the signal.

The final SNR is computed as

$$\text{SNR}(a_{\text{bi}}; \hat{a}_{\text{bi}}) = \frac{\mathbb{E}(a_{\text{bi}})}{\mathbb{E}(a_{\text{bi}} - \hat{a}_{\text{bi}})}. \tag{15}$$

## D    BETTER UNDERSTANDING OF $360°$ VIDEOS

We downloaded the $360°$ videos, necessary for the YT-Music dataset, directly from YouTube in the equirectangular projection format, under the `.webp` format, which allows the user to freely move the camera. However, this setup is not adequate for our pipeline, which uses visual information for conditioning the binaural generation. As such, we needed a method for capturing as much information from the provided video. We choose to trade off the quality of the image by applying a stereographic projection, which distorts the original image using a fisheye-like effect, but allows for capturing more information about the surrounding space, in the classical rectangular frame. For this operation, we set the horizontal and vertical field of view (hFOV and vFOV) to $300°$. During this transformation, we also set a sample aspect ratio (SAR) to 1, to ensure that pixels are kept as square as possible in the final output image. One example of such a transformation is depicted in Fig. 6.

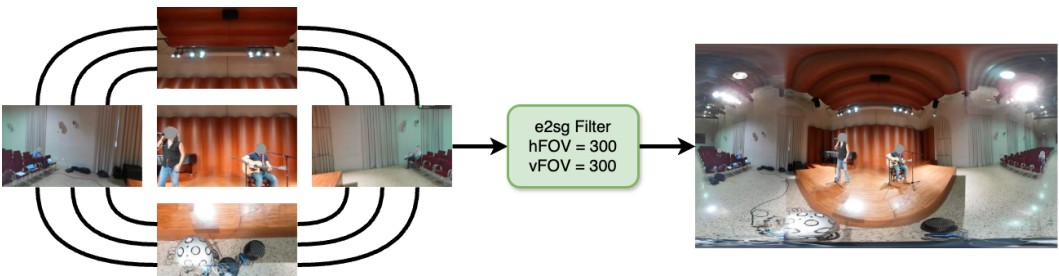

Figure 6: Equirectangular to stereographic projection

## E    COMPARISON WITH REAL AMBISONICS

Available datasets mostly contain the binaural audio under a simple stereo-like format, where only the left and right channels are provided. However, in the case of the YT-Music dataset, where the videos are $360°$, the audio also comes in different formats. One of them is the Ambisonics one, where some audio files are in the 3-channel $(W, X, Y)$ or 44-channel(includes $Z$) B-format of Ambisonics. Although we do not explicitly enforce the predicted channels, *i.e.* $\hat{X}, \hat{Y}$, Fig. 7 showcases similarities between predicted and real Ambisonics channels. Additionally, we present Table 4, in which we computed the STFT distance to numerically showcase the similarity between

the predicted and real Ambisonics channels. Since the magnitude level difference is not perceptible for the human ear, we also evaluated using the envelope distance (ENV), which requires the time-domain signal, smoothing the error during the inverse transform. As such, we see a low value for the ENV, indicating similar perceptual structure. Moreover, these results also emphasise the importance of the gain predictions for the Ambisonics FiLM layer, which are responsible for adjusting the magnitude of the predicted spectrograms.

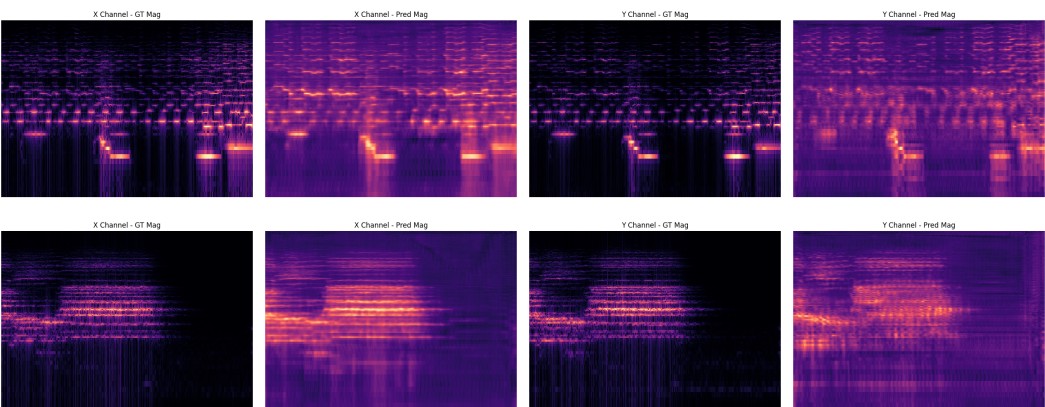

Figure 7: Visual analysis of real Ambisonics channels (first and third columns) and internally learned ones (second and last column). Please note that the model never saw real Ambisonics data. The predicted channels are taken before scaling and normalised for a better view.

| Methods | X channel | | Y channel | |
|---|---|---|---|---|
| | STFT $\downarrow$ | ENV $\downarrow$ | STFT $\downarrow$ | ENV $\downarrow$ |
| HiViBi | 5.084 | 0.006 | 1.779 | 0.002 |

Table 4: Metrics on the internally predicted Ambisonics channel.

## F  USER STUDY

In order to evaluate our proposed method in a subjective manner, we conducted a user study. The study was composed of a series of 20 videos, where the audio was obtained using different methods. We also included the ground truth video as a control mechanism. The participants were asked to rate each sample in terms of spatiality, with a value ranging from 1 (no spatiality) to 5, denoting a spatial audio. In total, 13 users with normal hearing participated in our study. From the results, presented in Fig. 8, we can conclude that the users considered our proposed solution to be the best one, excluding the ground truth, which further demonstrates the effectiveness of our method.

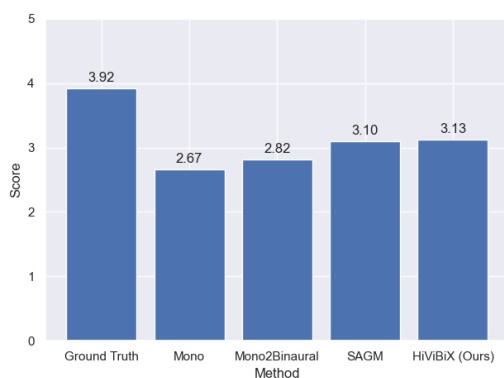

Figure 8: Results of the conducted user study. We asked the participants to rate videos containing binaural audio on a scale of 1 (worst) to 5 (best). Our method achieves the second position in terms of preferences, after the ground truth.

## G MORE EXAMPLES

In this section we showcase the results on the other two datasets. As such, Fig. 9 illustrates the results on the Music-Stereo dataset, while Fig. 10 is dedicated to the YT-Music one. For both figures, we focused on choosing examples that have a different data distribution from the FAIR-Play one to showcase their edge cases. On Music-Streore, we selected three examples, as follows: the first one is very focused on the people singing, with little room context; the second one provides an entirely different scenario, with the filming locations being outside, and the third example illustrates a case when one sounding instrument (guitar) is closer to the microphone than the other one (violin). We can see that in all cases, our proposed solution achieves impressive results, closely following the original signal.

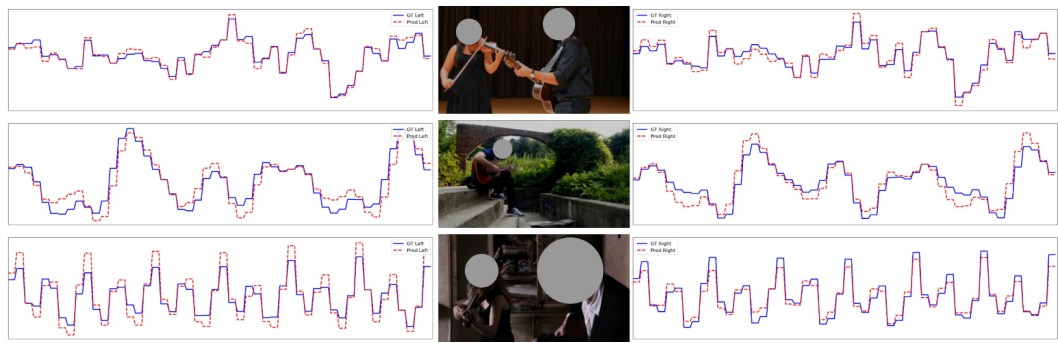

Figure 9: Examples of piecewise signals on Music-Stereo, ground truth (in blue) and predicted (in red) for the left and right channels.

For the YT-Music dataset, where the videos are 360, the model needs to have a larger understanding of the scene, as described in Appendix D. We selected examples which depict this phenomenon. On the first row, we have a crowded room with 5 different instruments, from multiple directions. The second row is dedicated to an outdoor example, while the third row showcases a music video, which can be catalogued as an outlier from the other examples.

**Failure cases.** Finally, we show one example in which our method is unable to grasp the full concept of the video. The example in Fig. 11 is taken from the YT-Music dataset and consists of a group of people dancing in a park. This dynamic video easily jailbreaks our proposed system, since we are only selecting one frame to extract the information from. As such, the full range of motions, and thus the acoustic propagation, might not captured enough in this scenario. As a solution, we propose two research directions: (i) estimating the best correlation/synchronization between a set of frames,

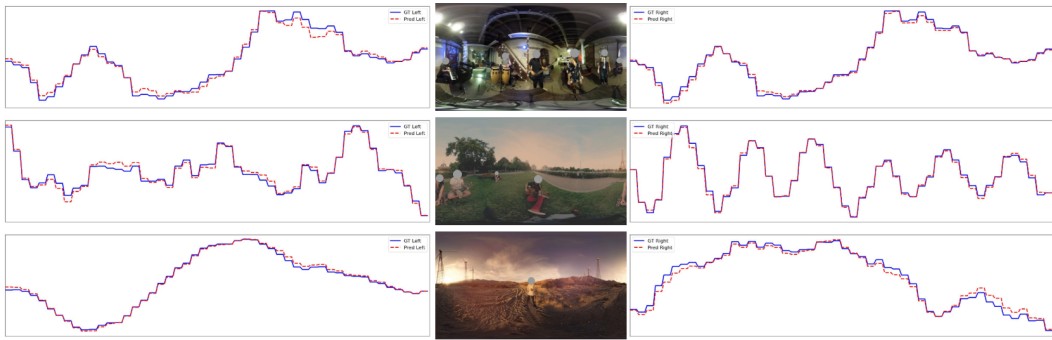

Figure 10: Examples of piecewise signals on YT-Music, ground truth (in blue) and predicted (in red) for the left and right channels.

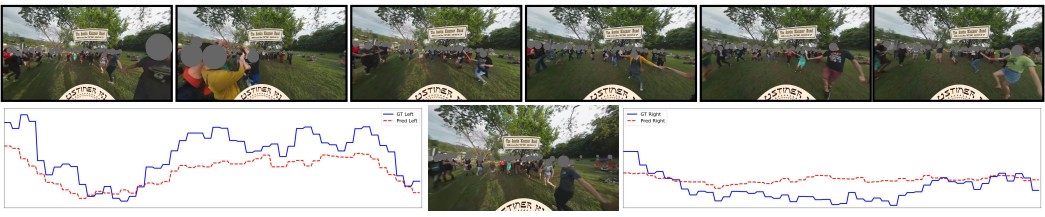

Figure 11: Example of a dynamic video: *people dancing*. The top part represents different frames from the video. The bottom part represents the left and right piecewise signals alongside the selected frame for analysis.

and the appropriate combination of the extracted features, and (ii) better motion representation of a dynamic video, *i.e.* moving sound sources, such as a car passing by.

## H  COMPARISON WITH GENERATIVE METHODS

We dedicate this section to comparing our method with other state-of-the-art solutions for video-to-spatial audio. Namely, we compare HiViBiX with ViSAGe (Kim et al., 2025) and OmniAudio (Liu et al., 2025) on their proposed datasets. Since these models aim at generating only coherent and plausible audio for the silent video, our distance-based metrics have a higher value for them. This phenomenon can also be seen in Fig. 12, where our method closely follows the ground-truth, because of the mono audio prior, while the OmniAudio variant is completely different.

| Methods | YT-Ambigen | | | Sphere360 | | |
|---|---|---|---|---|---|---|
| | STFT ↓ | ENV ↓ | SNR ↑ | STFT ↓ | ENV ↓ | SNR ↑ |
| Baseline (Mono-as-Stereo) | 2.953 | 0.244 | 0.000 | 3.926 | 0.260 | 0.000 |
| PseudoBinaural † *[CVPR '21]* | 2.096 | 0.257 | 4.371 | 3.383 | 0.250 | 1.045 |
| PseudoBinaural *[CVPR '21]* | 1.694 | 0.211 | 4.592 | 2.636 | 0.220 | 1.558 |
| CCStereo † *[ACM MM '25]* | 2.107 | 0.244 | 1.707 | 2.166 | 0.211 | 2.523 |
| CCStereo *[ACM MM '25]* | 1.813 | 1.716 | 4.208 | 2.064 | 0.207 | 6.350 |
| ViSAGe *[ICLR '25]* | 7.790 | 0.304 | -4.979 | - | - | - |
| OmniAudio *[ICML '25]* | - | - | - | 8.965 | 0.317 | -4.843 |
| **HiViBi** † (Ours) | 2.132 | 0.204 | 1.265 | 2.921 | 0.228 | 1.018 |
| **HiViBi** (Ours) | 0.997 | 0.147 | 5.804 | 1.010 | 0.164 | 3.916 |
| **HiViBi**-v2 (Ours) | 1.132 | 0.132 | 6.478 | 1.046 | 0.136 | 5.320 |

Table 5: Comparison of mono-to-binaural models against generative ones. The † signifies that the model is not trained on the specified dataset (zero-shot scenario). The v2 denotes changes made to HiVi encoder.

Table 5 reports the numerical results. For a fairer comparison, we evaluate PseudoBinaural (Xu et al., 2021) and CCStereo (Chen et al., 2025) both in a zero-shot setting (marked with †) and after fine-tuning them using their original training configurations, thereby establishing competitive baselines. For HiViBiX, we fine-tune starting from the YT-Music checkpoint. Since the two new datasets are not music-related, we also introduce a variant of the HiViBiX visual encoder, denoted v2. In this new variant, we replace YOLOv8 with the newer YOLOv11 and modify the detection strategy: instead of restricting detections to people, we use all available bounding boxes, irrespective of object category, *i.e.* all the possible 80 categories. We applied these modifications for v2 as an out-of-domain adaptation strategy, expanding the mono-to-binaural generation capabilities from music to more general audio. This leads to improved ENV distance and higher SNR on both datasets, at the cost of only a minor increase in STFT distance.

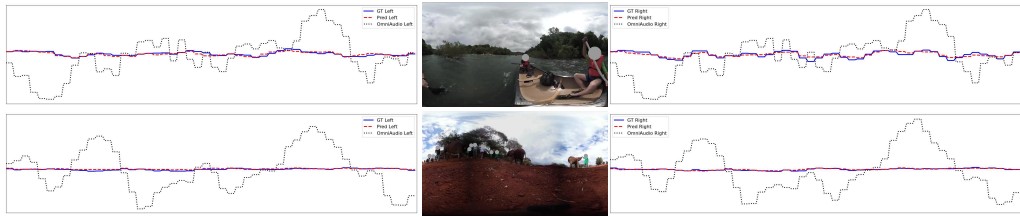

Figure 12: Piecewise signal comparison on Sphere360 dataset with OmniAudio.

