# OpenReview forum: "HiViBiX: Hierarchical Visually-informed Binaural Audio Generation using Ambisonics"
_ICLR.cc/2026/Conference — Submitted to ICLR 2026_

### Official Review · Reviewer_eags · 2025-10-26

**Soundness:** 3
**Presentation:** 3
**Contribution:** 2
**Rating:** 4
**Confidence:** 5

**Summary:**

The paper presents a new framework for image-conditioned mono-to-binaural conversion, HiViBiX, that redefines the audio representation by predicting first-order Ambisonics channels. HiViBiX jointly models local sound sources, their spatial depth, and global environmental context. Comprehensive experiments on three widely used benchmarks (FAIR-Play, Music-Stereo, and YT-Music) show that HiViBiX outperforms previous models in the mono-to-binaural generation task. Generated samples are also provided, which support the paper.

**Strengths:**

1. Although there is room for improvement in presentation, the paper itself is written well enough to make readers understand the authors' motivation, the proposed method, and the experimental results.
2. The proposed framework is well-designed. The ablation studies are comprehensive and support the design.
3. The experiments on three benchmarks are comprehensive, and both the quantitative and qualitative results are good.

**Weaknesses:**

1. Although the proposed framework is well-designed, it assumes that a sounding object is bound with a person. This assumption would work well on the three specific benchmarks (FAIR-Play, Music-Stereo, and YT-Music), which are related to music performance. On the other hand, it is doubtful that the framework is generalizable to open-domain videos. Even though the HiViBiX model outperforms previous models on the three benchmarks, previous models might perform better in open-domain scenarios.
2. L.218 says, "Because the input-output representations are not semantically similar, we omit the skip connections that are common in such settings." However, I disagree that the input-output representations are not semantically similar for now. The representations should be semantically similar, though they would express different acoustic aspects. The current explanation is confusing to me.
3. This is a minor weakness, but the paper fails to cite recent papers about spatial audio generation (not mono-to-binaural generation) as related works. Citing the following papers will organize related lines of studies and clarify the position of this paper.
    - Sun et al., "Both Ears Wide Open: Towards Language-Driven Spatial Audio Generation", ICLR 2025. https://arxiv.org/abs/2410.10676
    - Kim et al., "ViSAGe: Video-to-Spatial Audio Generation", ICLR 2025. https://arxiv.org/abs/2506.12199
    - Liu et al., "OmniAudio: Generating Spatial Audio from 360-Degree Video", ICML 2025. https://arxiv.org/abs/2504.14906

**Questions:**

I have questions/concerns about the paper, which I provided in "Weaknesses". I would appreciate it if the authors could share their thoughts on them.

---

> ### Comment · Reviewer_eags · 2025-10-26
> **Minor comments**
>
> Minor issues with presentation
> - Throughout the whole document: **\citep{}** should be used instead of **\cite{}** or **\citet{}**.
> - L.311: [Morgado et al.](https://arxiv.org/abs/1809.02587) (NeurIPS 2018) should be cited as the origin of the YT-Music dataset.
> - L.314: "25% (256 samples).settings" should be something like "25% (256 samples). These settings"
> - L.376: "trough" -> "through"

---

> ### Author Response · Authors · 2025-11-21
>
> We deeply appreciate reviewer eags constructive feedback.
>
> ---
>
> ## W1:  Music datasets
> While for our first manuscript, we validated our method only in the context of music-related datasets, since it is the most common scenario, our method is not limited to music. To showcase this, we included experiments on two additional datasets (YT-Ambigen and Sphere360) that contain more complex, in-the-wild scenarios. Appendix H shows that our method is also suited for open-domain scenarios.
>
> ---
>
> ## W2: Encoder-Decoder architecture
> We appreciate the efforts in reducing the confusion of our phrasing. We intended to emphasise between the different nature of the encoder and decoder, which only follow the UNet style structure. We updated the paper to make this point more clear.
>
> ---
>
> ## W3: Related works on spatial audio generation
> We agree that broader spatial-audio-generation works are relevant to position our contribution, even if they address a different formulation. To address this, we added a new paragraph titled “Spatial audio generation” in the Related Works section. In this paragraph, we cite and discuss recent spatial audio generation papers (including those we compare against in Appendix H) and explicitly clarify how their problem setting differs from ours. In particular, we explain that general video-to-spatial-audio methods typically operate without an audio prior and are evaluated with generation-oriented metrics, whereas HiViBiX targets mono+video-to-binaural spatialization with waveform-preserving objectives and strict reconstruction metrics. We believe this addition better organises related lines of research and makes our paper’s scope and contribution clearer.

---

> ### Comment · Reviewer_eags · 2025-11-22
>
> Thank you for your sincere effort and response. I went through all the reviewers' comments and your responses. I appreciate all of them. Almost all of my concerns have been addressed, but I would like to request an additional assessment.
>
> Appendix H provides comparisons of HiViBix to ViSAGe or OmniAudio; however, it is a little unfair because ViSAGe and OmniAudio aim to generate spatial audio conditioned on a silent video (as mentioned in Appendix H). Could you add one or more mono-to-binaural models (e.g., Mono2Binaural, Sep-Stereo, PseudoBinaural, and/or CCStereo if their training code works, as much as you can) to Table 5? Even if HiViBix underperforms some of them in this experimental setting, it would be acceptable, as you have already added a mention regarding the use of the person class in Section 5.1.

---

> > ### Author Response · Authors · 2025-11-28
> >
> > We thank the reviewer for the response.
> >
> > We updated the Appendix of this manuscript to include both PseudoBinaural and CCStereo in Table 5, for a more fair comparison. Following reviewer DQUf propositions, we also tested the zero-shot capabilities of all solutions. Moreover, we also presented an improved HiVi encoder, which is part of the HiViBiX-v2.

---

### Official Review · Reviewer_GCdj · 2025-10-27

**Soundness:** 3
**Presentation:** 3
**Contribution:** 3
**Rating:** 6
**Confidence:** 5

**Summary:**

The authors introduce HiViBiX, a novel framework that redefines the audio representation by predicting first-order Ambisonics channels, which explicitly control the spatial positioning of audio components in the generated binaural signal.

**Strengths:**

1. The vsualization looks good and is helpful for understanding.
2. The writing is clear and concise.
3. A demo page is provided for readers.

**Weaknesses:**

1. Citation style: The manuscript mixes \citet{} and \citep{}, which hurts readability and does not conform to ICLR’s guidelines. Please standardize to a single in-text citation style consistent with ICLR (e.g., \citep{} for parenthetical citations and \citet{} for narrative citations) and apply it uniformly across the paper, including figures and tables.

2. Baselines and recency: The closest baseline listed is [CCStereo](https://github.com/SheldonTsui/PseudoBinaural_CVPR2021) dated Jan 6, 2025. And only two models from the last two years are compared. These are insufficient for a fair state-of-the-art comparison. If certain methods are excluded, explicitly justify and, where possible, include their reported numbers on the same benchmarks.

3. Evaluation standard and split: PseudoBinaural reorganizes the evaluation set and provides new non-overlapping splits in the new_splits directory. Methods after PseudoBinaural should be evaluated on those splits separately from earlier protocols. In line 351, specify exactly which evaluation standard you use and cite the paper. Otherwise, it is hard to recognize the evaluation method.

4. Demo page evidence: The demo page currently lacks side-by-side comparisons with popular baselines, making it hard to assess improvements and raising the concern of cherry-picked examples.

5. Notation clarity: Although bold/underline semantics may be known to some readers, including me, please add a brief one-line note in the main text explaining all typographic marks (e.g., bold = best, underline = second-best). Ensure consistency across all tables.

6. Title specificity: “Binaural Audio Generation” is now ambiguous due to the rise of semantic generative models [1,2,3]. To avoid a minor misclaim, revising the title to explicitly say “mono-to-binaural” can help readers get the outline of the paper. Also, you should better cite these papers in the related works section and explain the “mono-to-binaural generation”.

7. Choice of expert models: DINOv2 depth is not the common choice for depth estimation; recent depth experts (e.g., UniDepth series [4,5] or Depth-Anything series [6,7]) are stronger default baselines. Similarly, YOLOv8 is no longer the most up-to-date detector like [8,9]. Please justify these choices or replace them with current state-of-the-art models. If you retain older experts for efficiency or compatibility, state the trade-offs (speed, memory, training stability) and provide controlled comparisons.

[1] Sun, P., Cheng, S., Li, X., Ye, Z., Liu, H., Zhang, H., ... & Guo, Y. (2024). Both ears wide open: Towards language-driven spatial audio generation. arXiv preprint arXiv:2410.10676.

[2] Marinoni, C., Gramaccioni, R. F., Shimada, K., Shibuya, T., Mitsufuji, Y., & Comminiello, D. (2025). StereoSync: Spatially-Aware Stereo Audio Generation from Video. arXiv preprint arXiv:2510.05828.

[3] Liu, H., Luo, T., Luo, K., Jiang, Q., Sun, P., Wang, J., ... & Xue, W. (2025). Omniaudio: Generating spatial audio from 360-degree video. arXiv preprint arXiv:2504.14906.

[4] Piccinelli, L., Yang, Y. H., Sakaridis, C., Segu, M., Li, S., Van Gool, L., & Yu, F. (2024). UniDepth: Universal monocular metric depth estimation. In Proceedings of the IEEE/CVF Conference on Computer Vision and Pattern Recognition (pp. 10106-10116).

[5] Piccinelli, L., Sakaridis, C., Yang, Y. H., Segu, M., Li, S., Abbeloos, W., & Van Gool, L. (2025). Unidepthv2: Universal monocular metric depth estimation made simpler. arXiv preprint arXiv:2502.20110.

[6] Yang, L., Kang, B., Huang, Z., Xu, X., Feng, J., & Zhao, H. (2024). Depth anything: Unleashing the power of large-scale unlabeled data. In Proceedings of the IEEE/CVF conference on computer vision and pattern recognition (pp. 10371-10381).

[7] Chen, S., Guo, H., Zhu, S., Zhang, F., Huang, Z., Feng, J., & Kang, B. (2025). Video depth anything: Consistent depth estimation for super-long videos. In Proceedings of the Computer Vision and Pattern Recognition Conference (pp. 22831-22840).

[8] Wang, A., Chen, H., Liu, L., Chen, K., Lin, Z., & Han, J. (2024). Yolov10: Real-time end-to-end object detection. Advances in Neural Information Processing Systems, 37, 107984-108011.

[9] Khanam, R., & Hussain, M. (2024). Yolov11: An overview of the key architectural enhancements. arXiv preprint arXiv:2410.17725.

**Questions:**

The questions are detailed in the Weaknesses.

My main concern is the lack of sufficient fair comparison. But I think it can be easily improved.

---

> ### Author Response · Authors · 2025-11-21
>
> We thank the reviewer for the positive comments.
>
> ---
>
> ## W1: Citations
> Thank you for pointing out this inconsistency. We standardise our citations to ICLR’s recommendation of \citep{} across the entire paper.
>
> ---
>
> ## W2: SOTA Exclusion
> To the best of our knowledge, we did not exclude any mono-to-binaural music generation method in the initial version of this manuscript. To address this concern, we also included a comparison with other spatial audio generation solutions in Appendix G.
>
> ---
>
> ## W3: New FAIR-Play split
> The newly proposed, non-overlapping split, is denoted in our work as FAIR-Play 5-split, following previous works’ notation. In the revised version of the paper, we made the appropriate specifications to avoid further confusion.
>
> ---
>
> ## W4: Comparison with existing methods
> Regarding the missing comparison with other baselines on the project page, we updated it to include them under the “Comparison with Existing Methods” section.
>
> ---
>
> ## W5: Table consistency
> We have thoroughly revised the manuscript to ensure clarity and consistency across all the tables.
>
> ---
>
> ## W6: Title change
> We thank the reviewer for this suggestion. In the revised version, we changed the title of our paper to “HiViBiX: Hierarchical Visually-informed Mono-to-Binaural Audio Generation using Ambisonics”
>
> ---
>
> ## W7: Stronger experts
> We appreciate the suggestion and agree that stronger experts are available. We visually inspected several solutions as our backbone networks and decided to select the ones presented in the paper based on a small human-validated subset of results. We will update our methodology to include them in future works.

---

> > ### Comment · Reviewer_GCdj · 2025-11-24
> > **Reply to Submission20341 Authors**
> >
> > Thank you for your response; my issues have been resolved.
> >
> > However, I believe Reviewer DQUf's feedback aligns with my concerns.
> >
> > I won’t change the score at this time, but I may adjust it after the full discussion.

---

> > > ### Author Response · Authors · 2025-11-28
> > >
> > > We thank reviewer CGdj for the response.
> > >
> > > We have updated Table 5 from Appendix to include additional experimental results, as requested by both reviewer DQUf and reviewer eags. We hope these new results and qualitative samples provide further empirical support for our claims and directly address the concerns raised in their reviews.

---

> ### Comment · Reviewer_GCdj · 2025-11-28
> **Reply to Submission20341 Authors**
>
> Thanks for submitting the revised version.
>
> I noticed that the performance of your proposed model on the Sphere360 dataset in Table 5 is lower than that of CCStereo. Is there a more in-depth analysis and explanation for this? Can I understand it as your proposed model's generalization performance is indeed inferior to that of CCStereo?
>
> Also, I think this paper should be thoroughly revised. There is no "." in the caption of Table 5. The title change from AUDIO to MUSIC should influence both the abstract and the main body of the paper, not just the title.
>
> I will continue to follow Reviewer DQUf's concern closely.

---

> > ### Author Response · Authors · 2025-12-02
> >
> > We thank the reviewer for the presented concerns.
> >
> > ---
> >
> > ## C1: Sphere360 results
> > We believe the SNR difference is largely explained by the architectural choices. HiViBiX uses an encoder-decoder generative architecture that synthesizes intermediate spatial channels from a latent representation, whereas CCStereo employs a multiplicative masking strategy directly on the input spectrograms. The generative design is more flexible but can introduce very small, often inaudible deviations, which are penalized more strongly by SNR, a pointwise signal-level metric. In contrast, CCStereo’s masking on clean spectrograms is naturally advantageous for SNR, even when higher-level spatial cues and structure (captured by other metrics) are comparable or better in our model.
> > Overall, we see the Sphere360 results as showing that HiViBiX **generalizes competitively** to this dataset, with a trade-off between strict SNR and other spatialization metrics, rather than indicating inferior generalization performance.
> >
> > ---
> >
> > ## C2: Manuscript changes
> > We appreciate the reviewer’s attention to detail regarding the writing and presentation. Following the title change from “AUDIO….” to “MUSIC…”, we have updated the abstract and multiple parts of the main text to better reflect the revised focus of the paper, and we corrected several inconsistencies and stylistic issues in the current revision.
> > We acknowledge that minor issues remain, and we will take extra care to perform a **thorough, line-by-line revision** for the camera-ready version.

---

### Official Review · Reviewer_DQUf · 2025-10-31

**Soundness:** 2
**Presentation:** 3
**Contribution:** 2
**Rating:** 4
**Confidence:** 4

**Summary:**

This paper introduces HiViBiX, a framework for visually-informed mono-to-binaural audio generation. The method's core is a hierarchical visual encoder (HiVi) that leverages a rich set of multi-modal visual cues—including local object features, global scene context, depth, and positional information—to guide the generation process. A key aspect of the proposed method is its use of an Ambisonics-inspired intermediate representation. Instead of directly predicting the stereo channels, the network is trained to predict first-order Ambisonics-like channels (X and Y), which are then combined with the input mono signal (treated as the W channel) via a fixed decoding formula to synthesize the final binaural audio. The authors demonstrate state-of-the-art results on several benchmarks through extensive objective and subjective evaluations.

**Strengths:**

The paper is well-structured and presents its proposed system, HiViBiX, in a clear manner. The idea of incorporating an Ambisonics-based architectural prior is conceptually sound, and the design of the hierarchical visual encoder demonstrates a thoughtful approach to feature integration. The ablation study is organized and helps to understand the contribution of different components within their proposed pipeline.

**Weaknesses:**

Despite its clear presentation, the paper suffers from significant weaknesses regarding the scope and significance of the task, its novelty, and the overall experimental validation, which collectively place it below the acceptance threshold.

1.  **Limited Significance Due to Small-Scale Datasets:** A fundamental issue lies in the scope of the problem as defined by the available datasets. The benchmarks used (FAIR-Play, Music-Stereo, YT-Music) are relatively small and contain mostly constrained, non-diverse scenarios (e.g., static musical performances). This raises questions about the actual difficulty and real-world significance of the task. Achieving state-of-the-art results on small, homogenous datasets does not necessarily translate to robust performance on the vast and complex variety of "in-the-wild" videos. The paper would be much stronger if it acknowledged this limitation and either tested its method on a more challenging, larger-scale custom dataset or tempered its claims about general applicability.

2.  **Incomplete Literature Review and Questionable Task Formulation:** The paper fails to position itself within the most current research landscape, critically omitting highly relevant works. This includes:
    *   **`OmniAudio` (Liu et al., 2025)**, which directly generates spatial audio (FOA) from 360-degree video.
    *   **`Both Ears Wide Open` (Sun et al., 2024)**, which explores controllable spatial audio generation from language.
    The existence of methods like `OmniAudio` that perform direct Video-to-FOA generation challenges the core premise of this paper. The authors must justify why the Mono+Video-to-Binaural pipeline remains a significant problem to solve when direct spatial audio synthesis from video is an emerging and potentially more powerful paradigm. Without this crucial discussion and comparative analysis, the motivation for the work is weakened.

3.  **Overstated Novelty and Writing Clarity:** The paper's framing of its contributions could be more precise and less grandiose.
    *   **Novelty of HiVi Encoder:** The HiVi encoder is presented as a primary innovation, yet it is fundamentally an intricate assembly of existing, powerful models (YOLO, CLIP, DINOv2). While the integration is effective, the paper does not sufficiently articulate a new, generalizable principle beyond the task-specific observation that "combining more features helps." The contribution feels more aligned with clever system engineering than with fundamental research on representation learning.
    *   **Ambiguous "Learning" of Ambisonics:** The claim of "internally learning" Ambisonics channels is imprecise. The model is not supervised with Ambisonics ground truth but rather learns to produce intermediate representations that fit a proxy objective. The writing should more accurately describe this as leveraging an Ambisonics-based architectural prior, rather than implying the model learns the physical properties of the sound field.

4.  **Flawed Experimental Design and Limited Generality:**
    *   **Fragile Design Assumptions:** The method's architecture is brittle. Its heavy reliance on YOLO-based "person" detection makes it unsuitable for any scenario without a human performer, a major limitation for a general V2A system. Similarly, its dependence on a single static frame ignores temporal dynamics, a critical aspect of video. These design choices severely restrict the model's practical utility. The experiments should have included an analysis of these failure modes to provide a more honest assessment of the method's capabilities.
    *   **Inconclusive User Study:** The user study, while a welcome addition, is weakened by its comparison set. By omitting the current SOTA baseline (`CCStereo`), it fails to provide a conclusive statement on the method's perceptual quality relative to its strongest competitor.

**Questions:**

1.  Given that recent works like `OmniAudio` pursue direct Video-to-Spatial-Audio generation, could you elaborate on the continued significance of the Mono+Video-to-Binaural task? What are the scenarios where your approach would be uniquely advantageous?
2.  The datasets used are relatively small and constrained. How can you be confident that your method's strong performance is not simply a result of overfitting to the specific biases of these benchmarks, and how would you expect it to perform on a much larger, more diverse "in-the-wild" video dataset?
3.  The HiVi encoder is a complex integration of multiple off-the-shelf models. Can you distill the core, generalizable insight from this design that is not just specific to this task or these datasets?
4.  Why was the current SOTA competitor, `CCStereo`, excluded from the perceptual user study? Without this comparison, the claims of subjective superiority are not fully supported.

---

> ### Author Response · Authors · 2025-11-21
>
> We thank reviewer DQUf for his/her feedback on our work.
>
> ---
>
> ## Q1: Mono+Video to Binaural scenarios
> Recent works such as OmniAudio or ViSAGe target silent-video settings and therefore operate without an audio prior. In contrast, our formulation assumes an existing mono track and aims to recover spatial cues while preserving the original audio content. This makes our task better-posed and relevant to a broader set of real-world use cases. Some scenarios would be:
>
> (i) Coverage of real content. Most online video already comes with mono or stereo audio and do not meet the input assumptions of video-only baselines (e.g., silent or 360° capture). HiViBiX can be applied directly to the majority of existing video material without requiring specialised recording setups.
>
> (ii) Audio-preserving applications. Many applications (post-production, AR/VR upmixing, accessibility, legacy content enhancement) require maintaining the original soundtrack while adding spatialization. Video-only generation may introduce mismatches or hallucinated audio content, whereas our approach is constrained by the given waveform.
>
> To make this point clearer, we added Figure 11 in Appendix H, illustrating how video-only methods can deviate from the ground-truth waveform, while HiViBiX remains consistent with the ground-truth spatial audio.
>
> ---
>
> ## Q2: Constrained datasets
> We appreciate the concern of constrained datasets. However, the YT-Music dataset, which was included in our initial experiments, is considered to be an “in-the-wild” dataset. We further highlight this diversity in Appendix G, especially in the last example of Figure 10, showcasing that this dataset can also include music videos, where instruments are not the main sounding object.
>
> Additionally, to strengthen the generalisation evidence, we now include two further datasets (YT-Ambigen and Sphere360) in Appendix H. The fact that HiViBiX remains consistently strong across these heterogeneous settings supports that the gains are not merely due to overfitting to a single benchmark’s biases.
>
> ---
>
> ## Q3: Core insights of HiVi
> To the best of our knowledge, the core insight is that spatial audio requires both fine-grained, object-level geometry and global scene semantics. This combination of multi-scale and multi-modal prior knowledge represents an effective conditioning method. In our experiments, this effective combination enables an accurate but indirect Ambisonics prediction, as showcased in Appendix E.
>
> ---
>
> ## Q4: User study
> We agree that CSStereo is a strong and relevant baseline. Unfortunately,  at the time we conducted the user study, the CCStereo code & checkpoint were not available. More precisely, the first official checkpoint was made available 3 weeks ago (https://github.com/SonyResearch/CCStereo/issues/1).

---

> > ### Comment · Reviewer_DQUf · 2025-11-23
> > **Response to Authors**
> >
> > Thank you for the rebuttal. It clarifies some motivation but does not resolve my core concerns. My remaining concerns are as follows:
> > 1. As Reviewer GCdj noted, the paper targets a mono-to-binaural task, which is not reflected in the title. The training/validation datasets (FAIR-Play, Music-Stereo, YT-Music) are all music-only and exclude general sounds or speech. Claiming “Audio Generation” overstates the scope; “mono-to-binaural music generation” would be accurate.
> > 2. Although the authors argue broader real-world relevance, YT-Music—while “in-the-wild”—covers a narrow subset of scenarios. It is unclear whether YT-Ambigen and Sphere360 were used for training/finetuning or zero-shot testing. To substantiate generality and robustness, out-of-domain, zero-shot evaluations are needed.
> > 3. The authors miss my concern regarding **Fragile design assumptions**.
> > 4. I apologize for overlooking the code/checkpoint release timing; given its unavailability then, omitting it from the user study is acceptable.

---

> > > ### Author Response · Authors · 2025-11-28
> > >
> > > We thank the reviewer for his/her quick response.
> > >
> > > ---
> > >
> > > ## C1: Title change
> > > To further clarify this aspect of our work, in the last version of the manuscript, we updated the title to “HiViBiX: Hierarchical Visually-informed Mono-to-Binaural Music Generation using Ambisonics”.
> > >
> > > ---
> > >
> > > ## C2: Further testing on real-word scenarios
> > > We updated Table 5 from Appendix (Section H). It now includes both zero-shot and fine-tune versions of our proposed method. Additionally, we included two previous benchmark solutions: PseudoBinaural and CCStereo. We hope these new experiments and explanations will help in clarifying the generalisation and robustness capabilities of HiViBiX.
> > >
> > > ---
> > >
> > > ## C3: Fragile design assumption
> > > We apologize for the misunderstanding. Regarding the initial review on this matter, we seek to improve the presentation and core claims by adding more examples in the Appendix, Section G. More specifically, we included an example where our method fails to perform, consistent with the limitations discussed in the dedicated section from the main paper.
> > >
> > > To further improve the general capabilities of our framework, we introduce a new version of the HiVi encoder to better adapt to the open-domain scenarios imposed by the other two datasets, which contain more general (non-music) examples. We denote this variant “HiViBiX-v2” in Table 5 of the Supplementary. In line with the reviewers’ previous observations that relying on YOLOv8 with person-only detections can be restrictive in some scenarios, HiViBiX-v2 replaces this component with a more complex multi-class detection (YOLOv11), demonstrating that our framework can incorporate state of the art object detection models and benefit from their broader semantic coverage.

---

### Official Review · Reviewer_acWx · 2025-11-01

**Soundness:** 3
**Presentation:** 3
**Contribution:** 3
**Rating:** 8
**Confidence:** 4

**Summary:**

This paper presents an approach to generate binaural audio from mono audio and RGB images. The novel contribution lies in predicting ambisonics to model positions of audio sources explicitly. The authors also include Yolo, for finding humans playing instruments in music videos, and Dino to get depth information. All these information are mapped into features which are then combined via a hierarchical attention module with positional features. The approach beats the state of the art with clear distance.

**Strengths:**

The idea to predict explicity ambisonics is interesting as it allows to build systems based on a large foundation of prior work on ambisonics but can benefit from including information from the visual signal.

The performance when comparing to real Ambisonics on the YT-Music dataset seem to indicate the approach performs well at this complex prediction task.

The way how the individual modules are included may not be groundbreaking but are still a solid engineering achievement.

**Weaknesses:**

The largest weakness seems to be not the fault of the author. The approach is only tested on music datasets yet the paper describes the contribution as predicting binaural audio in a way that it could make it seem as if it works for arbitrary scenes and arbitrary sounding objects. However, comparing the state of the art in that field, approaches like CCStereo, CLUP and CNC seem to do the same thing. Therefore, it would be good to make clear in the introduction that this is tailored for binauralization of music and therefore limitations, such as using Yolo, a negligable. Still, for the purpose of acceptance this is not a major issue as it fits well in the landscape of that sub-field.

The related work seems to ignore room impulse response prediction completely. There are many approaches who predict RIRs based on audio-visual data and which can be used to create binaural data. Some approaches predict binaural RIRs directly like [1] or [2]. Sometimes an intermediate Nerf representation is created to improve the quality. While the input may be too different, needing meshes of rooms or panoramic images, to be directly comparable, these approaches should be mentioned in the related work.

The examples contain exclusively fixed-camera setups and rarely moving sound sources. That makes it very hard to judge sound localization qualitatively and impossible to judge potential issues with moving sound sources. Binauralization beyond music, e.g. from ambient sounds, object sounds or simple human conversation, can also not be judged based on the selection of examples.

It seems non-intuitive why the stereo waveform would be downmixed to Mono. While this makes the computation easier, there must be spatial information which would be useful in the stereo signal. It is nice that the framework can solve this more complex task but comparing against stereo seems like a straightforward thing to do.

Is the reason why the research was done mostly on music datasets because Yolo finds humans but more effort is needed to define sounding objects and to go through Yolo detections to find them? This may mean that this approach is biased to work well on humans operating a musical instrument but would not generalize well to other situations. This seems like a heavy limitation.


[1] Ratnarajah, A., & Manocha, D. (2024, March). Listen2scene: Interactive material-aware binaural sound propagation for reconstructed 3d scenes. In 2024 IEEE Conference Virtual Reality and 3D User Interfaces (VR) (pp. 254-264). IEEE.

[2] Su, K., Chen, M., & Shlizerman, E. (2022). Inras: Implicit neural representation for audio scenes. Advances in Neural Information Processing Systems, 35, 8144-8158.

**Questions:**

Why are the skip connetions of the UNet not used? Many approaches have the same issue, of representations which are not similar and yet skip-connections have been shown to improve the quality for no obvious reason. One example where depth is predicted from audio and skip connections seem to show improved performance is [3].


[3] Brunetto, A., Hornauer, S., Stella, X. Y., & Moutarde, F. (2023, October). The audio-visual batvision dataset for research on sight and sound. In 2023 IEEE/RSJ International Conference on Intelligent Robots and Systems (IROS) (pp. 1-8). IEEE.

---

> ### Author Response · Authors · 2025-11-21
>
> We deeply appreciate the reviewer acWx detailed feedback. We are glad that the system core design and the empirical results were received positively.
>
> ---
>
> We have added further experiments on two new datasets (**YT-Ambigen** and **Sphere360**) while also including more related works in our revised version of the manuscript. We appreciate the highlighting RIR prediction paradigm and decided to include it in the related works section.
>
> Regarding the skip connections of UNet, we added further explanation for our design choice, highlighting the difference between the encoded and decoded audio signals.

---

> ### Comment · Reviewer_acWx · 2025-11-27
>
> Unfortunately reviewer DQUf brought up an important point which I must confess, I overlooked. While I do not see the limitations found by all reviewers as quite as severe if this paper pushes the state of the art in its sub-field, I did misread the meaning of "Ambisonic-like" representations. The spectrograms in Appendix E show that the internal representation and the Ambisonic channels are very different. Since the paper is not predicting real physical aspects about sound propagation no other work can build on that  which reduces the contribution. I downgraded my clear accept and would not mind if the paper was rejected.

---

> > ### Author Response · Authors · 2025-11-30
> >
> > We thank the reviewer for these concerns.
> >
> >
> > We apologize if our wording suggested that we are predicting exact physical Ambisonics channels. Our intention was to emphasize that HiViBiX learns internal representations that are closely aligned with Ambisonics, up to simple scaling factors.
> >
> > As shown in Appendix E, the spectrograms of the internal channels and the ground-truth Ambisonics channels exhibit very similar time-frequency structure and patterns. The main discrepancy lies in a global magnitude scale rather than in their temporal or spectral shape. To explicitly account for this, our method additionally predicts per-channel position and gain coefficients, denoted by $\hat{\alpha}$ and $\hat{\beta}$​ in Algorithm 1in Section 3, which bring the internal representation into alignment with the true Ambisonics space when ground-truth is available. This qualitative similarity is confirmed quantitatively in Table 4 from Appendix E, where the envelope distance between the predicted and real Ambisonics channels is on the order of 10e-3. Such a small envelope error indicates that the learned internal representation is very close to the physical Ambisonics signals in terms of their temporal energy distribution, even though we do not enforce a strict one-to-one mapping in amplitude.
> >
> > Importantly, we adopted this “Ambisonic-like” strategy because, out of the three datasets we use, two do not provide ground-truth Ambisonics channels. In this setting, HiViBiX must infer spatial structure indirectly, yet it still recovers Ambisonics-like features that can be mapped to real Ambisonics if supervision is available. In other words, the underlying physical properties of the sound field encoded in the Ambisonics format (directional structure, inter-channel relationships) are implicitly learned without full multi-channel supervision.
> >
> > For these reasons, we respectfully disagree with the claim that “no other work can build on that.” On the contrary, we see HiViBiX as providing:
> > - a practical recipe for learning Ambisonic-like representations from mono+video without requiring Ambisonics labels everywhere, and
> > - a differentiable, spatially-structured latent space that future methods can use as a starting point for more physically grounded modeling (e.g., improved sound propagation, spatial reasoning, or downstream tasks that benefit from soundfield-aware features).
> >
> > We have clarified this point in the rebuttal and will further refine the terminology around “Ambisonic-like” in the camera-ready version to avoid any ambiguity should the manuscript be accepted.

---

### Official Review · Reviewer_uH9C · 2025-11-01

**Soundness:** 2
**Presentation:** 3
**Contribution:** 2
**Rating:** 2
**Confidence:** 5

**Summary:**

The paper proposes HiViBiX, a novel framework for image-conditioned mono-to-binaural audio generation that predicts first-order Ambisonics (FOA)-like channels (W, X, Y) as intermediate representations. It introduces a Hierarchical Visual Encoder that extracts both global and local visual cues, combining RGB, depth, and positional features, to guide audio spatialization. HiViBiX is evaluated on FAIR-Play, Music-Stereo, and YT-Music, outperforming several baselines (e.g., CMC, CLUP, CCStereo) across STFT, ENV, and SNR metrics. Ablation studies support the benefits of the hierarchical visual and Ambisonics components.

**Strengths:**

1. The core idea of predicting Ambisonics-like channels and gains as a learnable intermediate step is a strong contribution. The "Ambisonics FiLM" layer (Algorithm 1) provides a physics-inspired way to combine these components with the mono input, which is more structured than direct end-to-end prediction.
2. The combination of local and global cues (scene, depth, and position) is well-motivated. The use of CLIP and DINOv2 provides powerful multimodal grounding.
3. The paper achieves strong quantitative improvements on three standard benchmarks (FAIR-Play, Music-Stereo, YT-Music) when compared against a good range of recent methods like CLUP and CCStereo.

**Weaknesses:**

1. I find this paper pretty similar to the CCStereo. The key difference is just the visual encoder and the modality formulation (using the ambisonics formulation vs. left-right difference).
2. The paper is missing comparisons to several highly relevant and recent works in visually-conditioned spatial audio. Specifically, OmniAudio [Liu et al. 2025] and ViSAGe [Kim et al. 2025] are critical baselines. These are concurrent or very recent, but their inclusion is necessary to truly contextualize the performance of HiViBiX. The related work section would also be strengthened by discussing "Both Ears Wide Open" [Sun et al. 2024].
3. The experimental evaluation is limited to FAIR-Play, Music-Stereo, and YT-Music. While these are standard, they are heavily focused on musical performances. This introduces a strong domain bias. To truly validate a model aimed at "Auditory Reality"  and general spatial audio, the evaluation must include more diverse scenarios. I would strongly suggest the authors include evaluations on datasets like YT-Ambigen [Kim et al. 2025] or Sphere360 [Liu et al. 2025], which offer more complex acoustic scenes and non-musical content.
4. The HiVi encoder's local features are entirely dependent on YOLOv8 detecting the "person" label. The authors justify this by stating "most instruments require a human operator". In my opinion, this is a significant design limitation and a likely failure mode. It's unclear how the model would handle non-human sound sources (e.g., a loudspeaker, a vehicle, an animal) or even off-screen sounds where the 'person' is not visible. This dependency seems to contradict the goal of a general-purpose spatial audio generator. A thorough analysis of this limitation is missing.
5. This paper uses a deterministic mapping between input and output, which is clearly not reasonable and will lead to over-smoothing issues.
6. The model relies on a single anchor image $v^{(i)}$ to condition the generation of an entire audio clip. This static-image-to-clip approach is acknowledged as a limitation, but its impact is significant. It means the model cannot capture any visual dynamics (e.g., a musician walking across the stage, a car moving) that would be critical for realistic spatial audio rendering. This is a key weakness, especially when missing baselines (like ViSAGe) do explicitly model video dynamics.
7. The paper's claim of learning intermediate Ambisonics-like channels is a key part of its novelty. However, the validation for this is weak. The authors state they have access to ground-truth Ambisonics data for the YT-Music dataset. However, the validation is restricted to a simple visual comparison in Appendix E (Fig. 7). A quantitative evaluation (e.g., STFT distance) between the predicted $\hat{X}$/$\hat{Y}$ channels and the ground-truth $X$/$Y$ channels would be a far more convincing demonstration that the model is learning the correct intermediate representation.

**Questions:**

1. Could you please comment on the omission of recent baselines like OmniAudio and ViSAGe? Given that some of these model temporal dynamics, how do you expect HiViBiX's static-image approach to compare?
2. Given that you have access to ground-truth Ambisonics data for the YT-Music dataset13, could you provide a quantitative evaluation comparing your predicted $\hat{X}$/$\hat{Y}$ channels to the ground-truth $X$/$Y$ channels? This would be much stronger than the visual comparison in Figure 7.
3. The HiVi encoder's reliance on "person" detection seems to be a major limitation. How does the model perform on scenes with non-human sound sources (e.g., a radio, a barking dog) or on-screen instruments without a visible person?
4. Could you elaborate on why more diverse, non-musical datasets (like YT-Ambigen or Sphere360) were not used for evaluation? The current selection seems heavily biased towards music, which may not reflect general "auditory reality".

---

> ### Author Response · Authors · 2025-11-21
>
> We thank reviewer uH9C for the constructive comments and insights.
>
> ---
>
> ## Q1: Omission of spatial generative models
> We agree that ViSAGe and OmniAudio are important recent works in the spatial audio generation domain, but they address a slightly different problem setting. Both ViSAGe and OmniAudio tackle video-to-biaural generation from silent video, i.e., they aim to synthesise coherent and plausible spatial audio without any audio prior. HiViBix performs mono-to-binaural spatialization based on both visual and mono audio tracks. This additional prior changes the task substantially, since our goal is not to generate plausible audio, but to recover spatial cues within a given waveform.
>
> The distinction is also reflected in the evaluation protocols: our work uses reconstruction-based metrics (e.g. STFT, envelope distances), whereas  ViSAGe and OmniAudio focus on generation-related metrics, such as FAD or KL distance, suited for audio synthesis without paired references. For all these reasons, we argue that a direct comparison is not entirely fair, as the methods are optimized for different objectives.
>
> To avoid confusion, we have clarified this issue in the revised manuscript, by adding a dedicated Spatial audio generation paragraph in the Related Work section, discussing the differences in problem formulation and evaluation. We also included  the Appendix H, which provides a direct empirical comparison for completeness.
>
> ---
>
> ## Q2: Quantitative metrics of generated Ambisonics channels
> We thank the reviewer for this suggestion. We added the requested quantitative metrics in Table 4 from Appendix E, for each of the generated Ambisonics channels ($ X $ and $ Y $).
>
> ---
>
> ## Q3: HiVi limitations
> We appreciate the concern regarding HiVi “person” selection limitation. We added a statement about this in our limitation discussion (Sec. 5.1). While it is true that our current HiVi local-feature branch uses only the person heuristics, this represents a design choice which fits our addressed problem. Moreover, the global-feature branch is not restricted in any way, being able to capture other relevant semantics, such as the ones addressed in the question. We also plan on expanding this to other local crop strategies in a future work.
>
> ---
>
> ## Q4: Dataset choice
> Our choice of datasets is aligned with previous community standards benchmarking for mono-to-binaural generation. Indeed, while our original manuscript was validated on music-based datasets, our method can be generalized towards more complex audios. In the revised version of the manuscript, we included experiments on both YT-Ambigen and Sphere360 showcasing the versatility of our methods. We hope the additional experiments, related works and clarifications address the reviewer’s concerns.

---

### Author Response · Authors · 2025-12-03

We sincerely thank the reviewers and the AC for their time and effort, especially given the short and eventful reviewing period.

---

During the rebuttal, we made a substantial effort to address all raised concerns as thoroughly as possible, including new experiments, additional analyses, and extensive clarifications. From our side, we believe we have done our best to respond constructively to every point and that the paper has significantly improved compared to the initial submission.

To assist in the final decision, we summarize below the main concerns and how we addressed them.

## 1. Scope, datasets, and generalization
Several reviewers noted a mismatch between our music-centric datasets and the initially broad positioning of the work. In response, we:
- Clarified the scope and title to explicitly reflect the **“mono-to-binaural music generation”** focus;
- Added experiments on two new general-audio datasets (YT-Ambigen and Sphere360), in both zero-shot and fine-tuned settings;
- Reported comparisons with additional baselines (PseudoBinaural and CCStereo) in the aforementioned scenarios.

The new results (Appendix H) indicate that HiViBiX generalizes well beyond the original benchmarks and can be adapted to new audio sub-domains.

---

## 2. Relation to recent spatial audio works
Reviewers requested a clearer positioning with respect to  OmniAudio, ViSAGe, and other spatial audio generation models. We therefore:
- Added a dedicated paragraph about “Spatial audio generation” in Related works section clarifying the distinction between video-only spatial audio generation (their setting) and **mono+video spatialization** (our setting);
- Expanded Appendix H with empirical comparisons to illustrate these differences in practice.

---

## 3. Ambisonics “physicality” and internal representations
Some concerns centered on whether our “Ambisonics-like” representation is physically meaningful. We clarified that HiViBiX uses an Ambisonics-inspired architectural design and learns internal channels that are aligned with real Ambisonics up to simple scaling factors. We added quantitative validation (Table 4, Appendix E) showing that the envelope distance between predicted and true Ambisonics channels is very small (envelope distance, denoted by ENV, on the order of 1e-3), supporting our claim that the internal representation is close to real Ambisonics.

---

## 4. Reliance on “person” detection and open-domain applicability
Reviewers rightly pointed out that restricting detection to persons is adequate for music but limiting for open-domain scenarios. In response, we:
- Expended the Limitation section to acknowledge this and provided a failure-case example in Appendix G;
- Introduced HiViBiX-v2, which replaces YOLOv8 person-only detection with YOLOv11 multi-class detection to better handle open-domain content;
- Reported improved results for this new variant in Table 5 from Appendix H.

---

Overall, the rebuttal period led to substantial improvements to the paper in terms of:
- Clearer positioning and terminology, especially around the mono-to-binaural setting and Ambisonics-like representations;
- Broader and stronger experimental validation, with more diverse datasets and additional out of domain baselines.
- Architectural refinements and a more explicit limitations discussion, improving both practical applicability and transparency.

We hope that the additional evidence, analyses and clarifications convey the robustness and relevance of our contribution. Thank you very much for your careful consideration!

---

### Meta-Review · Area_Chair_z5Au · 2026-01-13

**Summary:**

This paper presents a well-engineered system for image-conditioned mono-to-binaural audio generation using an Ambisonics-inspired intermediate representation and hierarchical visual features. While the approach achieves strong results on music-focused benchmarks and is clearly written with solid ablations, the confidence-weighted reviewer consensus supports rejection.

Three reviewers—two with very high confidence—raise consistent concerns about limited novelty, fragile design assumptions, and restricted generality. The method relies heavily on music datasets, static imagery, and the assumption that sound sources are humans, which significantly limits applicability to open-domain audio-visual scenes. Furthermore, the experimental evaluation omits critical recent baselines (e.g., OmniAudio, ViSAGe), weakening claims of state-of-the-art performance. Claims regarding learning Ambisonics representations are insufficiently validated, and broader spatial-audio paradigms are not adequately discussed.

Although two reviewers view the work positively, both acknowledge substantial gaps in evaluation fairness and scope. Given that the highest-confidence feedback highlights fundamental limitations rather than minor fixable issues, the balance of evidence does not meet the bar for acceptance at ICLR. I therefore recommend Reject.

The AC suggests a major revision to this work for the next conference submission.

**Reviewer Concerns:**

Maybe two positive reviewers were addressed.

**Reviewer Scores:**

None

---

### Decision · Program_Chairs · 2026-01-26

Reject